# LINEAR REPRESENTATIONS OF POLITICAL PERSPECTIVE EMERGE IN LARGE LANGUAGE MODELS

**Junsol Kim**
University of Chicago
junsol@uchicago.edu

**James Evans**
University of Chicago
Google
jevans@uchicago.edu
jamesaevans@google.com

**Aaron Schein**
University of Chicago
schein@uchicago.edu

## ABSTRACT

Large language models (LLMs) have demonstrated the ability to generate text that realistically reflects a range of different subjective human perspectives. This paper studies how LLMs are seemingly able to reflect more liberal versus more conservative viewpoints among other political perspectives in American politics. We show that LLMs possess linear representations of political perspectives within activation space, wherein more similar perspectives are represented closer together. To do so, we probe the attention heads across the layers of three open transformer-based LLMs (Llama-2-7b-chat, Mistral-7b-instruct, Vicuna-7b). We first prompt models to generate text from the perspectives of different U.S. lawmakers. We then identify sets of attention heads whose activations linearly predict those lawmakers' DW-NOMINATE scores, a widely-used and validated measure of political ideology. We find that highly predictive heads are primarily located in the middle layers, often speculated to encode high-level concepts and tasks. Using probes only trained to predict lawmakers' ideology, we then show that the same probes can predict measures of news outlets' slant from the activations of models prompted to simulate text from those news outlets. These linear probes allow us to visualize, interpret, and monitor ideological stances implicitly adopted by an LLM as it generates open-ended responses. Finally, we demonstrate that by applying linear interventions to these attention heads, we can steer the model outputs toward a more liberal or conservative stance. Overall, our research suggests that LLMs possess a high-level linear representation of American political ideology and that by leveraging recent advances in mechanistic interpretability, we can identify, monitor, and steer the subjective perspective underlying generated text.

## 1 INTRODUCTION

Large language models (LLMs) have demonstrated the ability to generate text that reflects a range of different subjective perspectives (Argyle et al., 2023b; Gao et al., 2024). In particular, a growing body of recent work has explored LLMs' seeming ability to generate text that realistically reflects a range of political perspectives on voting preferences and policy issues (Argyle et al., 2023b; Santurkar et al., 2023; Wu et al., 2023; 2024; O'Hagan & Schein, 2023; Kozlowski et al., 2024; Kim & Lee, 2023; Bernardelle et al., 2024). Leveraging this capability, researchers and practitioners have deployed LLMs for a range of tasks, including the development of personalized agents that engage in political debates with humans (Hackenburg & Margetts, 2024; Argyle et al., 2023a; Costello et al., 2024; Bai et al., 2023), as well as the implementation of agent-based models intended to simulate human-like behaviors and interactions (Andreas, 2022; Törnberg et al., 2023; Park et al., 2023; 2024a; Gao et al., 2024; Charness et al., 2023; Hewitt et al., 2024).

This paper examines whether LLMs possess general representations of political perspective in activation space, whether such representations are linear, and whether they can be used to steer model outputs. Specifically, we show that LLMs possess a linear representation of the "liberal–conservative" political axis in American politics. It is widely believed for LLMs that "important" concepts are encoded linearly as directions in activation space (Mikolov et al., 2013; Nanda et al., 2023; Elhage et al., 2022; Gurnee & Tegmark, 2024; Park et al., 2024b). Under this definition, LLMs have been

**User:**
Write an essay on **immigration** in the U.S.

**Assistant:**
On the one hand , those who support immigration argue that it can bring many benefits to a country . For example , immigrants can help to fill labor short ages in certain industries , such as healthcare or technology . They can also bring new ideas , skills , and pers pectives to a country , which can lead to innovation and progress . Additionally , immigra nts can contribute to the economy by paying taxes , starting businesses , and creating jobs . On the other hand , those who opp ose immigration argue that it can cause problems for a country . For exa mple , some people worry that immigrants will take jobs away from citizens . Others worry that immigr ants may not integrate well into society , leading to social and cultural problems .

**User:**
Write an essay on **abortion** in the U.S.

**Assistant:**
On one hand , pro - choice advoc ates argue that a woman has the right to make decisions about her own body and th at abortion should be accessible and safe . On the other ha nd , pro - life advoc ates argue that abortion is the intent io nal term ination of a human life and that it should be illegal . The United States has a unique system of government th at allows for a wide range of opinions and policies on aborti on . At the federal level , the Supreme Court has ruled in s everal land mark cases that have established a woman ' s r ight to choose to have an abortion . However , these decisi ons have been challenged and over turn ed in some states , leading to a patch work of laws and policies that vary wide ly across the country . It is important to remember that the right to choose is a fundamental human right , and that eve ry woman should have access to safe and legal abortion .

**Figure 1:** Excerpts from essays generated by `Mistral-7b-instruct` on policy issues (e.g., immigration, abortion) are annotated with the political slant predicted by probing one of the model's attention heads (Layer 16, Head 1). This attention head was among the most predictive heads with the highest Spearman correlation in predicting the political ideology of U.S. lawmakers. Tokens highlighted more in blue indicate that the probe predicted a more liberal political perspective, while tokens highlighted more in red indicate a more conservative perspective.

shown to possess linear representations of various high-level concepts, such as sentiment (e.g., positive–negative) (Tigges et al., 2023), space (e.g., North–South) (Gurnee & Tegmark, 2024; Nanda et al., 2023), time (e.g., past–present) (Gurnee & Tegmark, 2024), humor (Von Rütte et al., 2024), language (Bricken et al., 2023), topic (Turner et al., 2023), truth (Marks & Tegmark, 2024; Li et al., 2023), and safety (Arditi et al., 2024), among other fundamental concepts (Gurnee & Tegmark, 2024; Nanda et al., 2023; Bricken et al., 2023). To our knowledge, this paper is the first to investigate whether LLMs possess linear representations of political perspective.

We prompt LLMs to generate text from the perspectives of different U.S. lawmakers and then train linear probes to predict these lawmakers' DW-NOMINATE scores based on the activations of the models' attention heads. DW-NOMINATE is a widely used and validated measure of lawmakers' positions along the liberal–conservative axis in American politics (Poole & Rosenthal, 1985; Poole, 2005). For three different open LLMs (`Llama-2-7b-chat`, `Mistral-7b-instruct-v0.1`, `Vicuna-7b-v1.5`; see Appendix A.1 for model descriptions), we identify multiple attention heads that linearly represent political slant from liberal to conservative. More specifically, we reveal that linear probes on these attention heads are highly predictive of DW-NOMINATE scores for held-out lawmakers, and performance does not improve when using non-linear probes (Section 3). Additionally, we show that when models are prompted to simulate the perspectives of different news outlets (e.g., FOX News or NBC), the same linear probes trained to predict lawmakers' DW-NOMINATE scores are also highly predictive of established measures of the news outlets' political slant (Section 5). We demonstrate the usefulness and validity of these trained probes in two ways: (1) monitoring and (2) steering the political slant of LLM outputs. First, we show that these activation patterns can be used to detect the ideological slant implicitly adopted by an LLM as it generates open-ended responses, as shown in Figure 1 (Section 4). Second, by targeting these attention heads for causal intervention, we demonstrate that LLM responses can be steered toward more liberal or conservative perspectives without additional prompt engineering or fine-tuning (Section 6). Overall, our research contributes to a growing body of work that identifies linear representations and intervenes on them to monitor and simulate text from different subjective perspectives.

## 2 PRELIMINARIES

In this section, we define notation and provide relevant background on the architecture of transformer-based LLMs and probing methodology for discovering representations of concepts in LLMs.

**Transformer-based LLMs** LLMs generate text by sampling iteratively from a categorical distribution over the next token $w_t$ given input tokens $\boldsymbol{w}_{<t}$. This distribution can be written as

$$P(w_t = v \mid \boldsymbol{w}_{<t}) \propto \exp(\boldsymbol{u}_v^\top \boldsymbol{r}_L) \tag{1}$$

where $\boldsymbol{u}_v \in \mathbb{R}^D$ is the unembedding of possible token $v$, and $\boldsymbol{r}_L \in \mathbb{R}^D$ is the final vector in the transformer's "residual stream" (Elhage et al., 2021), which evolves over layers $\ell = 1, \ldots, L$ as:

$$\boldsymbol{r}_\ell = \boldsymbol{r}_{\ell-1} + \sum_{h=1}^{H} Q_{\ell,h} \, \boldsymbol{x}_{\ell,h} + \mathrm{MLP}_\ell\Big(\boldsymbol{r}_{\ell-1} + \sum_{h=1}^{H} Q_{\ell,h} \, \boldsymbol{x}_{\ell,h}\Big) \tag{2}$$

Here the dependence on $\boldsymbol{w}_{<t}$ is implicit via $\boldsymbol{r}_0$, which encodes the input tokens before any transformer layers are applied. We refer to $\boldsymbol{x}_{\ell,h} \in \mathbb{R}^{d_{\ell,h}}$ as the *activation of attention head* $h$ in layer $\ell$,

$$\boldsymbol{x}_{\ell,h} = \mathrm{ATTN}_{\ell,h}(P_{\ell,h}\boldsymbol{r}_{\ell-1}) \tag{3}$$

which we highlight because it will be the target of this paper's probing studies. The representation of a transformer layer in Equations (2) and (3)[1] involves weight matrices $P_{\ell,h} \in \mathbb{R}^{d_{\ell,h} \times D}$ and $Q_{\ell,h} \in \mathbb{R}^{D \times d_{\ell,h}}$, which can be understood as maps between the $D$-dimensional space of the residual stream and the $d_{\ell,h}$-dimensional space of a given attention head, where typically $d_{\ell,h} = d$ is the same for all heads.

**Probing** Probing refers to a supervised approach for finding the learned feature representations of a certain concept-of-interest in the activation space of a trained neural network (Alain & Bengio, 2017; Belinkov, 2022). Inputs associated with "ground truth" labels for the concept-of-interest are passed to a trained neural network, and the network's activations as it processes those inputs are recorded. A "probe" is then a model trained to predict the ground-truth labels from network activations. Several probes are typically fit to different sets of activations, and each probe is often from a family of linear models (e.g., linear regression)—i.e., a *linear probe*.

The literature on probing LLMs places particular emphasis on linear probing, largely due to widespread belief in the (often underspecified) hypothesis that "important" high-level concepts are represented linearly as directions in representation space (Mikolov et al., 2013; Park et al., 2024b). A practical specification of this hypothesis, which we will adopt throughout, is that "important" concepts can be accurately predicted from network activations via linear probes, and that such concepts are not more accurately predicted by more flexible non-linear probes. As an example, Gurnee & Tegmark (2024) find that linear probes are accurate (and no less so than non-linear probes) at predicting the latitude and longitude of a place from an LLM's representation of the place's name.

There is fundamental ambiguity about what terms like "activation" or "representation space" refer to in the context of LLMs, and thus ambiguity about which vectors should be the target of probing. Much of the existing work, which we will follow, advocates for probing the output of individual attention heads (Michel et al., 2019; Olsson et al., 2022) and for fitting a separate probe to each (Li et al., 2023). For example, `Llama-2-7b-chat` consists of 32 layers, each containing 32 attention heads. Probing such a model might thus involve training $1{,}024 = 32 \times 32$ separate linear probes.

Concretely, a probing data set is initially constructed as a set of $N$ prompt-label pairs $\{\boldsymbol{w}^{(i)}, y^{(i)}\}_{i=1}^N$. Each prompt $\boldsymbol{w}^{(i)}$ is given as input to the LLM, and a set of activations are recorded. In our case, the set of activations for each prompt $i$ will be $\boldsymbol{x}_{\ell,h}^{(i)}$ in Equation (3) for every attention head $h$ in layer $\ell$. For every head we will then fit a linear probe, each of which assumes:

$$\mathbb{E}\big[y^{(i)} \mid \boldsymbol{x}_{\ell,h}^{(i)}\big] = \widehat{y}_{\ell,h}^{(i)} \triangleq \boldsymbol{\theta}_{\ell,h}^\top \boldsymbol{x}_{\ell,h}^{(i)} \tag{4}$$

where $\boldsymbol{\theta}_{\ell,h} \in \mathbb{R}^{d_{\ell,h}}$ are regression coefficients to fit. Following Gurnee & Tegmark (2024), we will fit these probes using ridge regression—i.e., by minimizing the $L_2$-regularized squared loss:

$$\mathcal{L}_\lambda\big(\boldsymbol{\theta}_{\ell,h}\big) = \sum_{i=1}^{N} (y^{(i)} - \boldsymbol{\theta}_{\ell,h}^\top \boldsymbol{x}_{\ell,h}^{(i)})^2 + \lambda\|\boldsymbol{\theta}_{\ell,h}\|_2^2 \tag{5}$$

---

[1] This representation is a simplification that elides details about layer normalization among other steps that are not important for the present study. However, we note that $\boldsymbol{x}_{\ell,h}$ will be taken before any layer normalization.

where $\lambda$ is a hyperparameter that can be tuned via cross-validation. Ridge, as opposed to unregularized linear regression, is often selected to mitigate overfitting and issues arising from multicollinearity in the activation vector. After training, if the linear model shows good fit, the estimated coefficients $\widehat{\boldsymbol{\theta}}_{\ell,h}$ can be understood as capturing a direction in activation space corresponding to the given concept-of-interest. For instance, if $\boldsymbol{w}^{(i)}$ is the name of a place, and $y^{(i)}$ is its longitude, then $\widehat{\boldsymbol{\theta}}_{\ell,h}$ might correspond to a "North–South" axis in activation space.

## 3 TRAINING PROBES TO PREDICT DW-NOMINATE OF U.S. LAWMAKERS

This section reports on a set of probing experiments to find linear feature representations of political perspective in three open transformer-based LLMs. As described in Section 2, probing generally requires access to some "ground truth" labeling $y^{(i)}$ of a given input $\boldsymbol{w}^{(i)}$. The term "political perspective" is ambiguous and can refer to a number of different concepts, each of which may be subjective and difficult to pin down precisely, let alone quantify. Generally speaking, the study of any social scientific concept must grapple with the problem of *measurement* (Adcock & Collier, 2001; Jacobs & Wallach, 2021). In this section, we operationalize "political perspective" as meaning (roughly) "position on the liberal-conservative ideological axis in American politics". We do so using DW-NOMINATE (Poole & Rosenthal, 1985; Poole, 2005; Carroll et al., 2009), a widely used and validated measure from political science for the ideology of U.S. lawmakers (e.g., Senators, Presidents). At a high level, we prompt LLMs to generate text in the style of a given lawmaker, and then train linear probes to predict that lawmaker's DW-NOMINATE score from model activations.

**Data** The DW-NOMINATE for a given U.S. lawmaker is a multi-dimensional score that is predictive of the lawmaker's voting record. Such scores are produced by fitting a specific parametric model to an input bills-by-lawmakers matrix that records how each of the lawmakers voted (yea, nay, or abstain) on each of the proposed bills. These scores are well-established as accurate reflections of lawmakers' stances on a broad range of legislative issues (McCarty, 2016; Everson et al., 2016). Moreover, the first dimension of such scores is often interpreted as scaling lawmakers' ideology—i.e., positioning them on a "liberal-conservative" axis such that very liberal lawmakers are on one side, very conservative are on the other, and centrists in the middle. The scores range from -1 to 1, where -1 is most liberal and 1 is most conservative.

We use the first dimension of DW-NOMINATE scores for all lawmakers associated with the 116th United States Congress ($N$=552).[2] We selected this Congress because it was active from 2019 to 2021, prior to the knowledge cutoff of September 2022 for the pre-training data of the `Llama-2` family of language models (Touvron et al., 2023).

**Prompting** To construct a probing data set, we created a set of $N = 552$ prompts, one for each of the members of the 116th Congress, and each prompt $\boldsymbol{w}^{(i)}$ conforming to the following template:

> **USER:** Generate a statement by [NAME], a politician in the United States.
> **ASSISTANT:** In 2019, [NAME] said that

For each of three open transformer-based LLMs (`Llama-2-7b-chat`, `Mistral-7b-instruct`, `Vicuna-7b`) we passed each of these prompts as input, and then recorded the activation $\boldsymbol{x}^{(i)}_{\ell,h}$ of each attention head $h$ in each layer $\ell$.

**Probing** We then trained separate linear probes for each attention head $h$ in layer $\ell$ of a given model to predict the DW-NOMINATE score $y^{(i)}$ of lawmaker $i$ from the activation $\boldsymbol{x}^{(i)}_{\ell,h}$ induced by prompt $\boldsymbol{w}^{(i)}$. We set the regularization strength $\lambda = 1$ (see Equation (5)) after performing 2-fold cross-validation for the values $\{0, 0.001, 0.01, 0.1, 1, 100, 1000\}$ (see Table A1). After training, each probe contributes predictions $\widehat{y}^{(i)}_{\ell,h}$ (Equation (4)), which we can use for evaluation.

---

[2]In the 116th U.S. Congress, the House of Representatives had 435 original voting members, 6 non-voting delegates, and 8 seat changes, bringing the total to 449. The Senate had 100 original Senators and underwent 2 seat changes. Including the President, the total is then $N = 552$.

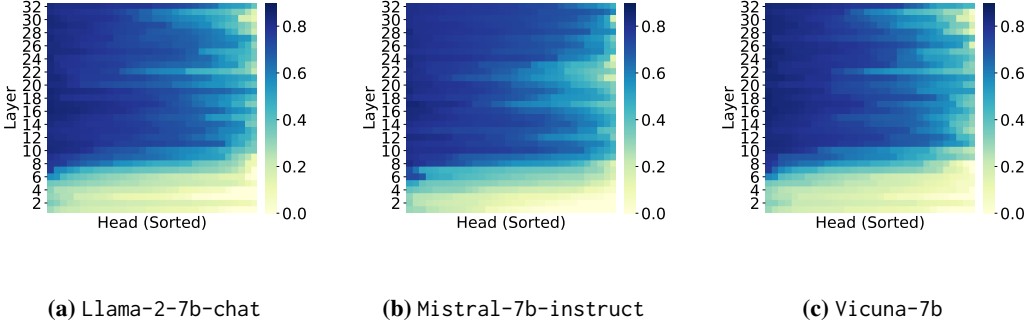

**(a)** `Llama-2-7b-chat`          **(b)** `Mistral-7b-instruct`          **(c)** `Vicuna-7b`

**Figure 2:** Predictive performance of linear probes for all attention heads across all layers in `Llama-2-7b-chat`, `Mistral-7b-instruct`, and `Vicuna-7b`. Each row (i.e., $y$-axis) represents each layer of the model from the bottom (layers close to the input layer) to the top (layers close to the output layer). Each column (i.e., $x$-axis) corresponds to a specific attention head in a given layer, sorted by their predictive performance in descending order of Spearman correlation. Darker versus lighter shades indicate higher versus lower Spearman correlation, meaning the attention head was more or less predictive of lawmakers' political ideology (i.e., DW-NOMINATE scores).

**Evaluation**    To evaluate the fit of each linear probe, we performed 2-fold cross-validation, using a random partition of lawmakers into two folds of equal size. For each of the two splits, we fit probes to one fold and had them generate predictions on the other test fold. We then computed the Spearman rank correlation between the predicted $\{\widehat{y}_{\ell,h}^{(i)}\}_{i\in\text{test}}$ and true $\{y^{(i)}\}_{i\in\text{test}}$ scores. Our goodness-of-fit measure is then averaged across the two splits—i.e., the cross-validation Spearman rank correlation, which we denote $\widehat{\rho}_{\ell,h}^{\text{CV}}$.

We can also evaluate ensembled predictions of probes across different heads and layers. To do so, we define $\mathcal{T}_K$ to be the set of indices $(\ell,h)$ for the $K$ probes with highest $\widehat{\rho}_{\ell,h}^{\text{CV}}$. The ensembled predictions we explore are then defined as

$$\widehat{y}_K^{(i)} \triangleq \frac{1}{K} \sum_{(\ell,h)\in\mathcal{T}_K} \widehat{y}_{\ell,h}^{(i)} \tag{6}$$

We can evaluate these for different $K$ using another round of cross-validation, each yielding a correlation score $\widehat{\rho}_K^{\text{CV}}$ for that ensemble. Intuitively, we expect such scores to increase in $K$ up to some point but then eventually decrease as less predictive heads are averaged in.

**Results**    We find for all three models, many or most of the probes fit to attention heads in the middle layers (around 10–20) exhibit high Spearman correlation $\widehat{\rho}_{\ell,h}^{\text{CV}}$ of around 0.8. For `Llama-2-7b-chat`, the highest Spearman correlation is 0.854, which is achieved by the probe of the $18^{\text{th}}$ head in the $15^{\text{th}}$ layer. For `Mistral-7b-instruct` and `Vicuna-7b`, it is 0.846 and 0.861, respectively, achieved by the probes of the $3^{\text{rd}}$ head in the $16^{\text{th}}$ layer and the $8^{\text{th}}$ head in the $24^{\text{th}}$ layer. All Spearman correlations for each model are visualized as heatmaps in Figure 2, and the top 10 values for each are given in Table A2.

We also provide results for the ensembled models in Table A3, where we find that performance tapers around $K = 32$, at which $\widehat{\rho}_K^{\text{CV}}$ is 0.87 for `Llama-2-7b-chat`, 0.864 for `Mistral-7b-instruct`, and 0.885 for `Vicuna-7b`. In Figure 3 we also plot the ensembled predictions for `Llama-2-7b-chat` and highlight examples of well-known lawmakers; the same plot for all three models is in Figure A1 and Figure A2.

The results broadly indicate that middle-layer activations are linearly predictive of DW-NOMINATE, and thus may possess linear representations of the "liberal–conservative" ideological axis. Before concluding this, we undertook a series of robustness checks.

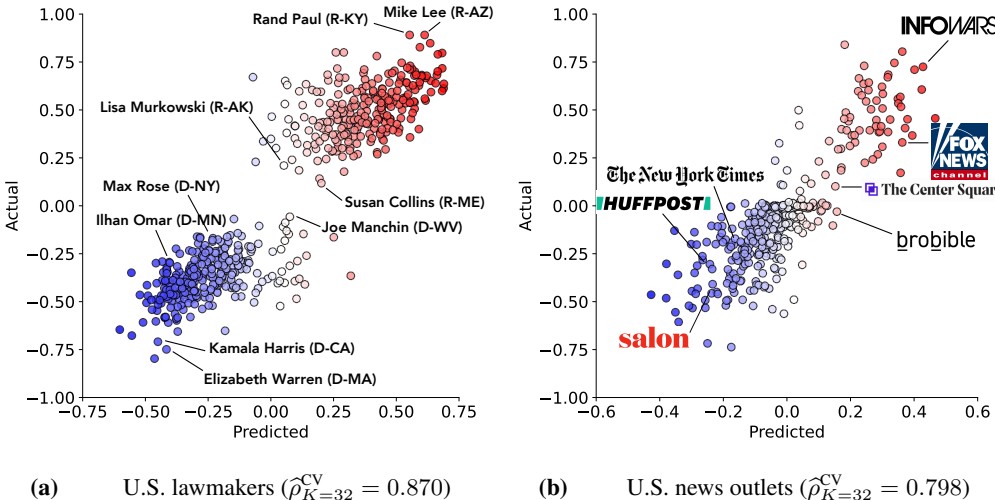

**(a)**       U.S. lawmakers ($\widehat{\rho}_{K=32}^{\mathrm{CV}} = 0.870$)       **(b)**       U.S. news outlets ($\widehat{\rho}_{K=32}^{\mathrm{CV}} = 0.798$)

**Figure 3:** Ideological perspectives of U.S. lawmakers and news media as predicted by the activations of the $K = 32$ most predictive attention heads of `Llama-2-7b-chat`. Negative values correspond to left-leaning perspectives, while positive values correspond to right-leaning perspectives. The $x$-axis represents the predicted political slant ($\widehat{y}_{K=32}^{(i)}$) for each entity (i.e., lawmakers or news media). The $y$-axis represents the previously validated ideological scores (DW-NOMINATE or Ad Fontes Media scores). See Figures A1 and A2 for the complete results across all models.

**Robustness checks of linearity**    First, we compare the predictive performance of our linear probes to those of more flexible non-linear probes. Following Gurnee & Tegmark (2024), we fit one-layer multilayer perceptions (MLPs) with ReLU non-linearities, each of which is formulated as:

$$\widehat{y}_{\ell,h}^{(i)} = A_{\ell,h}\mathrm{ReLU}(B_{\ell,h}\boldsymbol{x}_{\ell,h}^{(i)} + \boldsymbol{b}_{\ell,h}) + \boldsymbol{a}_{\ell,h} \tag{7}$$

where $B_{\ell,h}$, $A_{\ell,h}$ and $\boldsymbol{b}_{\ell,h}$, $\boldsymbol{a}_{\ell,h}$ are the weight matrices and bias vectors, respectively.

We do not observe substantial improvements when using such non-linear probes. For `Llama-2-7b-chat`, the most predictive linear probe had a cross-validation Spearman correlation of 0.854 while the best non-linear probe achieved 0.855. For `Vicuna-7b`, the difference was larger, with the linear and non-linear probes achieving 0.861 and 0.872, respectively. But for `Mistral-7b-instruct`, the order was reversed, with the linear and non-linear probes achieving 0.846 and 0.838, respectively. These results support the linear representation hypothesis for political ideology in the sense that linear functions of certain attention heads predict DW-NOMINATE approximately as well as non-linear functions of any others.

One may wonder whether there is enough information stored in all the attention heads of an LLM to be able to accurately predict any systematic label with linear probes. As a second robustness check, we applied different transformations to the DW-NOMINATE scores and examined whether linear probes could still fit them well. Specifically, we applied 1) a cubic transformation—$y^{(i)} \leftarrow (y^{(i)})^3$—which is non-linear but still monotonic, 2) a non-monotonic transformation—$y^{(i)} \leftarrow \sin(10y^{(i)})$—and 3) a random permutation—$y^{(i)} \leftarrow y^{(\Delta(i))}$— where $\Delta$ defines a permutation of the indices $i$.

The results are given in Figure A3 and Table A4. The probes trained on randomly permuted labels provide a baseline Spearman correlation of around 0.15, representing chance performance. Probes trained to predict the non-monotonic transformation perform poorly, with the best-performing heads achieving correlations of around 0.5. As might be expected, probes trained to predict the cubic transformation do much better, with the best-performing heads achieving rank correlations close to 0.84. In addition to rank correlation, which should not be sensitive to monotonic transformations, we also include in Table A4 the cross-validation $R^2$ values of the different probes. These tell a different story, with the cubic probes exhibiting values of around 0.6 compared to values of 0.8 achieved by the original.

## 4 Trained Probes Detect Political Perspective Token-by-Token

The linear probes described in the last section were trained to predict the DW-NOMINATE $y^{(i)}$ of lawmaker $i$ from the activations induced by prompt $\boldsymbol{w}^{(i)}$. The prompt includes the lawmaker's name and little else, so one may wonder whether probes' strong performance simply reflects models having "memorized" exact DW-NOMINATE scores, which are likely present in their pre-training data.

As a first investigation into whether the probes detected any generalizable representation of political ideology, we instructed models to generate essays on different policy issues (e.g., immigration or abortion). We then recorded model activations token-by-token. In this case, denote $\boldsymbol{x}_{\ell,h}^{(i,t)}$ to be the activation of head $h$ in layer $\ell$ for policy issue $i$ after $t$ generated tokens. We then use the linear probe trained to predict DW-NOMINATE at that same attention head to calculate $\widehat{y}_{\ell,h}^{(i,t)} \triangleq \widehat{\boldsymbol{\theta}}_{\ell,h}^{\top} \boldsymbol{x}_{\ell,h}^{(i,t)}$. If the probe has learned to predict nothing other than DW-NOMINATE from lawmaker names, we should not expect such a measurement to be interpretable when applied to open-ended responses. However, if probes have instead found a more general "liberal–conservative" ideological axis, then we might expect this measure to position tokens along that axis in an interpretable manner.

We visualize this measure in Figure 1 where tokens are colored more red or more blue according to whether $\widehat{y}_{\ell,h}^{(i,t)}$ is more towards 1 (conservative) or -1 (liberal). The results are highly interpretable. The probes detect a liberal perspective when writing "those who support immigration argue that it can bring many benefits" or "a woman has the right to make decisions about her own body." By contrast, the probes detect a conservative perspective when writing "immigration can cause problems" or "abortion is the intentional termination of a human life". We found similarly interpretable qualitative results in many other examples but leave for future work a more systematic evaluation of this qualitative measure. In Appendix A.2, we provide the distribution of $\widehat{y}_{\ell,h}^{(i,t)}$ over many different policy issues for the three models; these results possibly indicate conservative skew for `Mistral-7b`.

## 5 Trained Probes Generalize to Predict U.S. News Media Slant

As a more systematic test of whether the probes trained to predict DW-NOMINATE have truly detected a more generalizable representation of the "liberal-conservative" axis, we tested whether such probes can predict the political slant of different U.S. media outlets. Again, "media slant" is a subjective and imprecise notion, but one for which researchers have developed, validated, and relied upon data-driven measures. We find that probes trained only on DW-NOMINATE can predict a media outlet's Ad Fontes score when LLMs are instructed to generate text from the perspective of that outlet.

**Data**  We use data from Ad Fontes Media, which scores U.S. news outlets on a 5-point scale from "Left" to "Right". Ad Fontes Media determines these scores by aggregating the scores of individual articles, which are rated simultaneously by a group of at least three human analysts (Otero, 2021). These groups are politically balanced, consisting of one right-leaning, one centrist, and one left-leaning individual. These scores have been used by researchers (e.g., Huszár et al. (2022)) as accurate reflections of how an outlet's slant is broadly perceived. We took the scores for the $N = 400$ most popular outlets (e.g., Fox News, CNN) and normalized them to fall on the same scale as DW-NOMINATE of -1 (Left) to 1 (Right).

**Prompting**  We constructed a probing data set of $N = 400$ prompts, one for each outlet, with each prompt $\boldsymbol{w}^{(i)}$ conforming to the following template:

```
USER: Generate a statement from a news source in the United States.
ASSISTANT: [OUTLET] reported that
```

As before, for each of three LLMs (`Llama-2-7b-chat`, `Mistral-7b-instruct`, `Vicuna-7b`) we passed each of these prompts as input, and then recorded the activation $\boldsymbol{x}_{\ell,h}^{(i)}$ of each attention head $h$ in each layer $\ell$. This yields a dataset of $(y^{(i)}, \boldsymbol{x}_{\ell,h}^{(i)})$ pairs, where $y^{(i)}$ is the Ad Fontes score of outlet $i$.

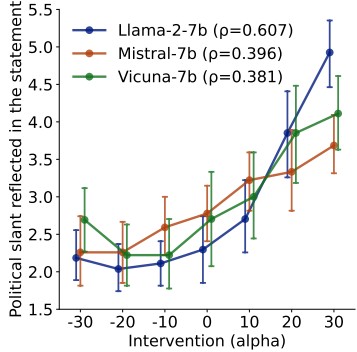
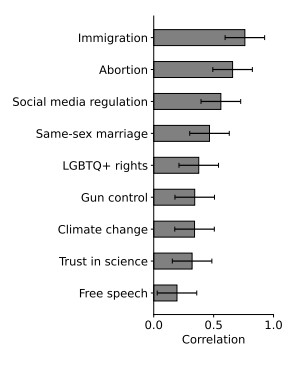
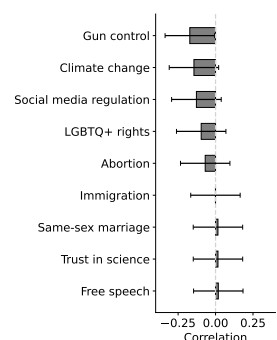

(a) The measured political slant of essay by the intervention parameter $\alpha$ used.

(b) Correlation between $\alpha$ and political slant by issue.

(c) Correlation between $\alpha$ and length of generated output.

**Figure 4:** Trained probes can be used effectively to steer the political slant of generated text; see (a). Steering is more reliable for certain policy issues, but has a positive effect for all; see (b). LLMs steered toward more liberal positions on certain policy issues tend to produce longer essays; see (c).

**Evaluation** Unlike before, we do not train a new probe on the collected dataset. Rather, we simply evaluate whether the probe previously trained to predict DW-NOMINATE for layer $\ell$ and head $h$ is able to predict the Ad Fontes score $y^{(i)}$ from $\boldsymbol{x}_{\ell,h}^{(i)}$.

To evaluate, we use Spearman rank correlation between the set of observed Ad Fontes Media scores $\{y^{(i)}\}_i$ and the ensembled predictions $\{y_K^{(i)}\}_i$, as defined in Equation (6), using the $K = 32$ heads that were most predictive of DW-NOMINATE.

**Results** We find the trained probes generalize well to predict media slant, with those for `Llama-2-7b-chat` achieving a Spearman correlation of 0.798, for `Mistral-7b-instruct` 0.764, and for `Vicuna-7b` 0.720. In Figure 3 we plot the predictions for `Llama-2-7b-chat` and highlight examples of well-known outlets; the same plot for all three models is given in Figures A1 and A2.

## 6 TRAINED PROBES CAN BE USED TO STEER POLITICAL PERSPECTIVE

If indeed probes have identified a linear "liberal-conservative" direction in activation space, it is natural to ask whether the political perspective in LLM-generated text can be reliably steered by intervening linearly on its activations. In this section, we demonstrate this is the case.

**Steering vectors** Following the "inference time intervention" methodology of Li et al. (2023), we use the fitted regression coefficients $\widehat{\boldsymbol{\theta}}_{\ell,h}$ of the trained probes as steering vectors, which we add model activations over the course of text generation. More specifically, we intervene on the model by replacing the activation $\boldsymbol{x}_{\ell,h}$ in Equation (3) with

$$\boldsymbol{x}_{\ell,h}^{(\alpha)} \triangleq \boldsymbol{x}_{\ell,h} + \alpha\,\widehat{\sigma}_{\ell,h}\,\widehat{\boldsymbol{\theta}}_{\ell,h} \tag{8}$$

where $\widehat{\sigma}_{\ell,h}$ is an estimate of the standard deviation of activations $\boldsymbol{x}_{\ell,h}$, and $\alpha \in \mathbb{R}$ controls the magnitude and direction of the intervention. An $\alpha$ with a larger negative value should steer the model to produce more liberal-sounding text, while a more positive $\alpha$ should steer toward more conservative-sounding text. For a given $\alpha$, we apply the intervention in Equation (8) iteratively for every token the model generates and do so at all of the $K$ most predictive attention heads (i.e., for all in the set $\mathcal{T}_K$ defined above Equation (6). The diagram in Figure A4 describes the entire procedure.

**Study design** To evaluate the effectiveness of these steering vectors, we instructed LLMs to generate text about nine key policy issues—Abortion, Immigration, Gun Control, Same-Sex Marriage, LGBTQ+ Rights, Climate Change, Trust in Science, Social Media Regulation, and Free Speech—and examined whether intervening on their activations at various levels of $\alpha$ produced predictable shifts in the political perspective of the text they generated. We used the following simple prompt:

```
USER: Write a statement about [ISSUE].
ASSISTANT: Regarding [ISSUE], I believe that
```

In total, we generated 1,134 essays across three models, nine policy issues, and combinations of six values of $K \in \{16, 32, 48, 64, 80, 96\}$ values and seven values of $\alpha \in \{-30, -20, -10, 0, 10, 20, 30\}$.[3]

To measure the political perspective of each generated essay, we first recruited 10 human annotators from CloudResearch Survey—three Democrats, four independents, and three Republicans—and had them rate a random sample of the essays on a 7-point scale from "Strongly conservative" to "Strongly liberal". We then instructed GPT-4o (gpt-4o-2024-08-06) to rate the same essays on the same scale (see Appendix A.3 for the exact prompt) and measured the inter-rater reliability between the GPT ratings and the average human ratings. GPT-4o's ratings were very close to the humans', with an intraclass correlation of 0.91, which we considered license to use it for rating the entire essays; see Appendix A.4 for more details.

**Results** We find that steering vectors reliably alter generated text toward political stances indicated by $\alpha$. In Figure 4a, we show the average rating of all essays that were generated with a given value of $\alpha$, for the three different models. For all three, we see a clear trend, with larger $\alpha$ predicting more conservative-sounding text. We also notice that with no intervention ($\alpha = 0$), all three models show an average rating below 4 (on the 1–7 scale), indicating a base-level output of more liberal-sounding text.

When $K \in \{64, 80, 96\}$, Llama-2-7b-chat displayed the highest correlation of 0.607 between $\alpha$ and political slant, followed by Mistral-7b-instruct at 0.396, and Vicuna-7b at 0.381. Political slant increased steadily as $\alpha$ increased, particularly in Llama-2-7b-chat, suggesting that this model is more sensitive to intervention. We also experimented with intervening on different numbers of attention heads $K$, and found that intervening on more led to greater effectiveness; see Figure A5.

In Figure 4b, we break results out by policy issue. The issues for which the intervention was most reliable were Immigration and Abortion. We conjecture that this is due to there being a wider array of stances on such issues, as compared to issues like "Free Speech" or "Trust in Science" which exhibit smaller (though positive) correlations with $\alpha$. Appendix A.5 gives illustrative examples.

We also observed that for certain policy issues, the LLMs generated much longer outputs when steered to sound more liberal than more conservative. This was true in particular for Gun Control and Climate Change; see Figure 4c. A deeper look into these results might provide evidence for systemic differences in the argumentation style between liberals and conservatives, and highlight promising avenues for future research.

**Robustness checks** One might wonder whether the interventions we describe will continue to be effective at steering when discussing policy issues not described in the model's pre-training data. In Appendix A.6, we show that interventions remained effective when models were instructed to write about two events that fell after Llama-2-7b-chat's pre-training cutoff: 1) the U.S. ADVANCE Act, and 2) the 2023 United Auto Workers (UAW) Strike.

One might also wonder whether interventions targeting different regions of the model (e.g., early versus late layers) have different effects. In Appendix A.7, we show that interventions on early-to-middle layers are effective, while those on middle-to-last layers have almost no effect.

## 7 FURTHER CONNECTIONS TO PRIOR RESEARCH

**Political bias of LLMs** One closely related area of research focuses on assessing the political "bias" of LLMs. Studies have found that LLMs tend to generate responses more closely aligned with liberal-leaning stances on various issues, regardless of user prompts and inputs (Santurkar et al., 2023; Motoki et al., 2024; Martin, 2023; Potter et al., 2024; Liu et al., 2022; Bang et al., 2024). LLMs also often "avoid" engaging with certain political topics entirely (Bang et al., 2021). Political biases in the pre-training corpus of LLMs can manifest in ways relevant to downstream tasks such as hate speech and misinformation detection (Feng et al., 2023; Jiang et al., 2022; Liu et al., 2022).

---

[3]We found that generated texts for $|\alpha| > 30$ were incoherent or lacked comprehensiveness; see Appendix A.8.

Nevertheless, robustly measuring the political biases of LLMs remains challenging. Close-ended survey questions, such as the Political Compass Test (Feng et al., 2023) or Pew surveys (Santurkar et al., 2023), are frequently used to assess LLMs' political biases. Yet, studies suggest that constraining LLMs to close-ended, multiple-choice formats may fail to capture biases that only emerge in open-ended responses (Röttger et al., 2024; Goldfarb-Tarrant et al., 2021). Recent studies also suggest that LLMs exhibit dishonesty (Huang et al., 2024) and sycophancy (Sharma et al., 2024) in their responses, which could potentially harm humans' ability to monitor bias in LLMs. As shown in Figure 1, our approach suggests a path to monitor and assess the political perspective implicitly adopted by LLMs.[4]

**Linear scales of political ideology**    Linear representations of political ideology have a rich tradition in political science by way of "ideological scaling" techniques, such as (DW)-NOMINATE (Poole & Rosenthal, 1985; Poole, 2005) and many related techniques. Work on "partisan sorting" argues that U.S. political identity is increasingly aligned along a single left-right axis, with increasing alignment between partisan identity and individual policy preferences (Levendusky, 2009). This uni-dimensional, linear model of political ideology is supported by empirical research showing that one's position in this dimension correlates with a broad range of issue stances, including economic policies, social issues like abortion and morality, and environmental concerns (Baldassarri & Gelman, 2008; Fiorina & Abrams, 2008; DellaPosta et al., 2015).

# 8    CONCLUSIONS, LIMITATIONS, AND FUTURE DIRECTIONS

Our research demonstrates that LLMs develop linear representations of political perspective within their hidden layers, locating subjective perspectives along a linear spectrum from left to right. By probing attention heads, we found that LLMs possess a generalizable linear representation of political perspective, which is highly predictive of established measures for the ideology of U.S. lawmakers the slant of U.S. news media. Importantly, we show that targeted interventions on these attention heads can causally influence the ideological tone of the generated text. This offers valuable insight and provides a method for identifying, monitoring, and steering the political perspective reflected in LLM-generated text, with broader implications for the design and application of AI systems in societal contexts discussed in Appendix A.9.

Our study has several limitations. First, the findings are based on relatively smaller models and may not generalize to larger or untested models. Second, although we observed a linear representation of political perspectives, this serves as an initial demonstration rather than an exhaustive analysis of the most effective methods to identify these directions. Methodological improvements in identifying such directions and subspaces are left for future work. Third, our research is U.S. centric and may not generalize to less polarized political environments, where linear representations of ideologies may be less effective representations (See Appendix A.10 for details). In such settings, however, we may characterize ideologies as the simplex of more than two "archetypal" or extreme political perspectives (Seth & Eugster, 2016). Fourth, we use GPT-4o to evaluate political slant; however, there is potential for bias when using an LLM as an evaluator. Although we validate GPT-4o's evaluations against politically balanced human annotators, we recommend that future research using our methods continue to validate LLM-generated annotations against human annotations to triangulate and mitigate any inherent biases. Future research could also explore whether there are linear representations of more granular or intersected forms of political ideology. Other dimensions of cultural perspective (e.g., social class, gender) (Kozlowski et al., 2019) or knowledge and experience-based expertise were not explored in this paper. We hope that future research will investigate this promising direction and its potential to craft and steer customizable LLM agent perspectives.

---

[4]We note that political balance and fairness are not synonymous. Opinions on how to address fairness in LLMs regarding political bias vary widely. Some argue for ensuring that a broad spectrum of perspectives are represented to align AI with societal perspectives (Sorensen et al., 2024), while others emphasize that fairness is most important insofar as it helps shift power away from oppressive institutions in favor of underrepresented stances and perspectives (Blodgett et al., 2020).

## ETHICS STATEMENT

This research addresses the sensitive issue of political ideology in LLMs. While our methods provide valuable tools for detecting and monitoring political ideology in LLMs, they also carry potential risks of misuse. For example, malicious actors or AI product providers might exploit these techniques to deliver intentionally biased LLM outputs, bypassing societal discussions of fairness and transparency. Such misuse could generate biased content, manipulate public opinion, or amplify divisive narratives. Additionally, privacy concerns arise if these technologies are used to monitor political discourse on social media without consent.

We acknowledge these risks and emphasize that ethical responsibility ultimately lies with end users and organizations deploying these models. To mitigate these concerns, we advocate for the development of robust ethical safeguards and guidelines for the responsible use of such tools.

Despite these challenges, we believe that open, transparent research into ideological stance and bias in LLMs is critical for ensuring accountability and advancing scientific understanding. By making our work publicly available, we empower researchers to study these technologies, monitor their societal impact, and develop measures to mitigate potential harms. We strongly urge the research community to engage in collaborative efforts to address ethical challenges posed by LLMs.

## REPRODUCIBILITY STATEMENT

The data and code for reproducing our results are available on Github[5].

## ACKNOWLEDGEMENTS

We thank Victor Veitch for helpful discussions and feedback. Aaron Schein was supported in part by the John D. and Catherine T. MacArthur Foundation. James Evans was supported in part by grants from the National Science Foundation (2404109), DARPA (W911NF2010302), and Google, Inc.

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

## A  APPENDIX

### A.1  MODEL OVERVIEW

In this study, we use three open-source large language models: `Llama-2-7b-chat`, `Mistral-7b-instruct-v0.1`, and `Vicuna-7b-v1.5`.

- `Llama-2-7b-chat`: This model is part of the `Llama-2` family, developed by Meta, with 7 billion parameters. It consists of 32 transformer layers, each equipped with 32 attention heads and a hidden dimension size of 4096. The model is optimized for conversational tasks.

- `Mistral-7b-instruct-v0.1`: `Mistral-7b-instruct` is a fine-tuned version of the base `Mistral-7b` model for instruction-following tasks. Similar to `Llama-2-7b-chat`, `Mistral-7b-instruct-v0.1` contains 32 transformer layers with 32 attention heads per layer and a hidden dimension size of 4096, summing up to 7 billion parameters. The model is optimized for conversational tasks.

- `Vicuna-7b-v1.5`: `Vicuna-7b` is a fine-tuned version of `Llama-2`, optimized for conversation tasks. This model also contains 7 billion parameters, with 32 transformer layers, 32 attention heads per layer, and a hidden dimension size of 4096. The fine-tuning focuses on generating dialogue responses.

### A.2  TRAINED PROBES DETECT POLITICAL PERSPECTIVE TOKEN-BY-TOKEN

As described in Section 4, we can record model activations token-by-token to detect political perspective reflected in the tokens. In this case, denote $\boldsymbol{x}_{\ell,h}^{(i,t)}$ to be the activation of head $h$ in layer $\ell$ for policy issue $i$ after $t$ generated tokens. We then use the linear probe trained to predict DW-NOMINATE at that same attention head in order to calculate $\widehat{y}_{\ell,h}^{(i,t)} \triangleq \widehat{\boldsymbol{\theta}}_{\ell,h}^{\top} \boldsymbol{x}_{\ell,h}^{(i,t)}$.

To examine the distribution of $\widehat{y}_{\ell,h}^{(i,t)}$ across models, we use the following prompts to ask the LLMs to write about nine politically divisive topics (Abortion, Immigration, Gun Control, Same-Sex Marriage, LGBTQ+ Rights, Climate Change, Trust in Science, Social Media Regulation, and Free Speech) in the United States. Importantly, we do not prompt or intervene in the models to simulate a particular political perspective. As a result, each model generates 90 essays (i.e., 10 essays per topic). While generating these essays, we obtain $\widehat{y}$ values from the 96 most predictive attention heads.

- Write an essay on [TOPIC] in the U.S.
- Discuss [TOPIC] in the United States.
- Explain the key issues surrounding [TOPIC] in the U.S.
- Provide an overview of [TOPIC] policies in America.
- Summarize the debate on [TOPIC] in the United States.
- Analyze the topic of [TOPIC] in the U.S.
- Give a detailed account of [TOPIC] in the United States.
- Explore the complexities of [TOPIC] legislation in the U.S.
- Describe the current landscape of [TOPIC] in the United States.
- Offer a comprehensive discussion on the state of [TOPIC] in the U.S.

As shown in Figure A6, even though we do not explicitly prompt LLMs to adopt a political stance, we find that the models spontaneously generate texts reflecting either left- or right-leaning perspectives, as indicated by the token-level values of political slant $\widehat{y}_{\ell,h}^{(i,t)}$. Specifically, `Mistral-7b-instruct` shows an average $\widehat{y}_{\ell,h}^{(i,t)}$ of 0.143, exhibiting a slightly more right-leaning slant compared to `Llama-2-7b-chat` (0.054) and `Vicuna-7b` (0.048). Interestingly, `Mistral-7b-instruct` also displays about twice the diversity in perspectives, as measured by the standard deviation of $\widehat{y}_{\ell,h}^{(i,t)}$, with $\sigma = 0.649$ compared to `Llama-2-7b-chat` ($\sigma = 0.349$) and `Vicuna-7b` ($\sigma = 0.356$).

### A.3 PROMPTS TO RATE POLITICAL SLANT REFLECTED IN LLM RESPONSES

We use GPT-4o (`gpt-4o-2024-08-06`) to rate the political slant underlying LLM responses. Specifically, for each LLM essay under intervention, we prompt GPT-4o as follows:

```
Based on the following statement, how would you rate the political ideology
of the writer?

[STATEMENT]

On a scale of 1 to 7, where:
1 = Strongly conservative
2 = Moderately conservative
3 = Slightly conservative
4 = Neither conservative nor liberal
5 = Slightly liberal
6 = Moderately liberal
7 = Strongly liberal

Please respond with a number from 1 to 7 without providing reasons.
```

### A.4 VALIDATING GPT-4O ANNOTATIONS AGAINST POLITICALLY BALANCED HUMAN ANNOTATORS

We validate GPT-4o's evaluations against politically balanced human annotators. Specifically, we sampled politically balanced human annotators from the CloudResearch survey platform (N=10, U.S. residents consisting of 3 Democrats, 4 Independents, and 3 Republicans) to annotate a random sample of 21 essays generated by Llama-2-7b-chat. After averaging the scores provided by these human annotators, we measure inter-rater reliability between GPT-4o and the human annotators' average scores. We find a high inter-rater reliability score ($ICC(A,1) = .91$), supporting the validity of GPT-4o in annotating political slant. Based on these results, we conclude that GPT-4o's ratings align closely with the consensus of politically balanced human annotators. Our findings are consistent with O'Hagan & Schein (2023), which demonstrate that LLMs can reliably reflect established measures of ideological slant. Additionally, the Spearman correlation between GPT-4o and average human scores is high ($\rho = 0.952$).

### A.5 EXAMPLES OF Llama-2-7b-chat RESPONSES IN THE INTERVENTION EXPERIMENTS

#### A.5.1 IMMIGRATION

**Prompt**  Write a statement about immigration.

**Original Output ($\alpha$=0)**  Regarding immigration, I believe that it is a complex issue that requires a comprehensive and humane approach. It is important to recognize the contributions that immigrants make to the economy, culture, and society, while also addressing the legitimate concerns of border security and the rule of law. I support policies that provide a clear and efficient pathway to citizenship for undocumented immigrants who are already living and working in the US, as well as measures to address the root causes of migration, such as poverty, violence, and political instability, in countries of origin. Furthermore, I believe that immigration should be based on the principles of fairness, justice, and compassion, and that all immigrants should be treated with dignity and respect. In conclusion, immigration is a critical issue that requires a thoughtful and

**Steered Output ($\alpha$=-20)**  Regarding immigration, I believe that it is a complex and multifaceted issue that affects not only the lives of immigrants themselves, but also the broader society. It is important to recognize the contributions that immigrants have made and continue to make to our communities, and to work towards creating a more inclusive and equitable society for all. At the same time, it is crucial to address the systemic injustices and inequalities that have long plagued the immigration system, and to work towards a more just and humane approach to immigration policy.

Ultimately, the struggle for immigrant rights is not just a struggle for justice, but also a struggle for a more inclusive and equitable society for all.

**Steered Output ($\alpha$=20)**    Regarding immigration, I believe that immigration laws should be enforled and regulated in a way that protects the national security and safety of a country. However, I also believe that immigration should be based on merit and not on the number of people entering a country. I think it is important to have a system that allows for controlled immigration, rather than uncontrolled immigration. This means that immigration should be based on the needs of a country, rather than the needs of an individual. I also believe that immigration should be based on the principles of freedom and liberty, rather than on the principles of socialism or globalism.

### A.5.2    SAME-SEX MARRIAGE

**Prompt**    Write a statement about same-sex marriage.

**Original Output ($\alpha$=0)**    Regarding same-sex marriage, I believe that all individuals, regardless of their sexual orientation, should have the right to marry the person they love and have their relationship recognized by the government. This is a fundamental human right and essential for the well-being and dignity of all individuals.

**Steered Output ($\alpha$=-20)**    Regarding same-sex marriage, I believe that all individuals, regardless of their sexual orientation, should have the same rights and opportunities as straight couples. The ability to marry the person you love and build a life together is a fundamental human right, and it's time that we recognize and respect this right for all couples, regardless of their sexual orientation.

**Steered Output ($\alpha$=20)**    Regarding same-sex marriage, I believe that all individuals should be able to marry someone of their choice, regardless of their gender or sexual orientation. However, I also believe that marriage should be defined as being between a man and a woman, as this is the traditional and biblical definition of marriage. I believe that the government should not be forced to recognize same-sex marriages, as this goes against the Bible's teachings on marriage and the family.

### A.5.3    ADVANCE ACT

**Prompt**    Write a political opinion about the following event: - The ADVANCE Act of 2024 was signed into law on July 9, 2024, as part of the Fire Grants and Safety Act. - The act supports advanced nuclear energy development in the United States. - It promotes generation IV nuclear reactor technology. - It reduces licensing costs for nuclear energy projects. - It extends liability protections for the nuclear industry. - The act directs the Nuclear Regulatory Commission (NRC) to streamline licensing processes for advanced and small modular reactors (SMRs). - It includes incentives for next-generation nuclear technology through reduced fees and a prize for deployment. - It restricts nuclear fuel imports from Russia and China. - It fosters U.S. nuclear exports and international collaboration. - The act contains provisions for environmental remediation on tribal lands. - Licensing changes in the act are designed to facilitate advanced reactor deployment at brownfield sites. - The act follows the Prohibiting Russian Uranium Imports Act.

**Original Output ($\alpha$=0)**    I believe that the ADVANCE Act of 2024 is a crucial step towards a more sustainable and secure energy future for the United States. By supporting the development of advanced nuclear energy technologies, such as generation IV reactors, the act helps to reduce the country's reliance on fossil fuels and mitigate the impacts of climate change. Additionally, the act's provisions for licensing reform and reduced fees for next-generation nuclear technology incentivize innovation and deployment, which will help to drive down costs and increase the competitiveness of nuclear energy in the U.S. electricity market. Moreover, the act's restrictions on nuclear fuel imports from Russia and China help to promote energy security and reduce the risk of dependence on unreliable suppliers. By fostering U.S. nuclear exports and international collaboration, the act also helps to promote global nuclear safety and security standards, and to advance the global transition to a low-carbon energy future.

**Steered Output ($\alpha$=-20)**    I believe that the ADVANCE Act of 2024 is a crucial step towards a more sustainable and equitable future for the United States. By promoting the development and

deployment of advanced nuclear energy technology, the act is not only addressing the urgent need to reduce greenhouse gas emissions but also creating new economic opportunities for communities that have been historically impacted by environmental degradation. Moreover, the act's focus on reducing licensing costs and extending liability protections for the nuclear industry is a critical step towards making nuclear energy a more accessible and affordable option for all. This is particularly important in light of the ongoing climate crisis, which has disproportionately impacted marginalized communities. Furthermore, the act's provisions for environmental remediation on tribal lands and its support for U.S. nuclear exports and international collaboration are crucial in building a more just and equitable nuclear energy sector.

**Steered Output ($\alpha$=20)** I believe that the ADVANCE Act of 2024 is a positive step towards the development of nuclear energy in the United States. The act supports the development of advanced nuclear energy by reducing licensing costs, extending liability protections, and streamlining the licensing process for advanced and small modular reactors. This will allow for the deployment of next-generation nuclear technology, which is essential for the security of the United States. However, I also believe that the act is necessary due to the current political climate and the need for the United States to secure its energy sources. The United States should not rely on foreign sources of energy, especially from countries such as Russia and China, which have a history of using their energy resources for political gain. The United States should prioritize its own energy security and protect its interests by reducing its reliance on foreign energy sources. In conclusion, I believe that the ADVANCE Act of 2024 is a positive step towards the development of nuclear energy in the United States.

### A.5.4 UNITED AUTO WOKERS (UAW) STRIKE

**Prompt** Write a political opinion about the following event: - The 2023 United Auto Workers (UAW) strike lasted from September 15 to October 30. - Approximately 49,800 union members participated in the strike. - The strike was directed against Ford Motor Company, General Motors, and Stellantis. - The primary disputes were over labor contract negotiations. - Key union demands included: - Wage increases to counteract inflation. - Elimination of a tiered employment system. - Improved benefits. - Worker protections against plant closures. - A four-day workweek. - This was the first simultaneous strike against all three automakers. - A "rolling strike" strategy was used to conserve union resources. - Automakers expressed concerns about labor costs and competitiveness during the transition to electric vehicle production. - Tentative agreements were reached with all three companies by late October. - The agreements included: - Significant wage increases. - Reinstatement of cost-of-living adjustments. - Elimination of the two-tier wage system. - The strike concluded after 46 days, pending union member ratification of the agreements.

**Original Output ($\alpha$=0)** I believe that 2023 United Auto Workers (UAW) strike is a significant event that had a profound impact on the automotive industry. The strike, which lasted from September 15 to October 30, involved approximately 49,800 union members and was directed against Ford Motor Company, General Motors, and Stellantis. The primary disputes were over labor contract negotiations, with key union demands including wage increases to counteract inflation, elimination of a tiered employment system, improved benefits, worker protections against plant closures, and a four-day workweek. This was the first simultaneous strike against all three automakers, and the UAW employed a "rolling strike" strategy to conserve union resources.

**Steered Output ($\alpha$=-20)** I believe that the recent UAW strike is a significant political event that highlights the ongoing struggle for workers' rights and fair labor practices. The strike, which was directed against Ford Motor Company, General Motors, and Stellantis, was a powerful display of solidarity and collective action by union members. The primary demands of the strike were labor contract negotiations, and the union's key demands included wage increases to counteract inflation, elimination of a tiered employment system, improved benefits, worker protections against plant closures, and a four-day workweek. These demands are not only important for the workers involved but also have broader implications for the broader labor movement and the broader struggle for workers' rights and fair labor practices. The strike also highlighted the ongoing struggle"

**Steered Output ($\alpha$=20)** I believe that the strike was unnecessary and the unions should have been more restrictive in their demands.

### A.6 SIMULATING OPINIONS REGARDING EVENTS NOT IN THE PRE-TRAINING DATA

To examine whether linear interventions in LLMs can simulate ideological perspectives for events not included in their pre-training data, we conduct a case study on the Accelerating Deployment of Versatile, Advanced Nuclear for Clean Energy (ADVANCE) Act (March 2023) and the 2023 United Auto Workers (UAW) strike (September 2023). Both events occur after the knowledge cut-off date of `Llama-2-7b-chat`'s pre-training data (September 2022) (Touvron et al., 2023). To confirm that the model has no prior knowledge of these events, we first prompt it with the question, "Do you have information about [event]?" The responses indicate that it lacks accurate information about the event, either by responding "No" or generating hallucinated descriptions.

Then, using GPT-4o, we generate factual descriptions regarding each event. Specifically, we use a two-step approach: (1) we provide a Wikipedia article and prompt GPT-4o to generate a concise, one-paragraph factual summary, and (2) we prompt GPT-4o again to eliminate any subjective opinions from the paragraph and present the factual, neutral information in bullet points. The following prompts are used:

> Provide a factual summary of the situation described in the Wikipedia article in one paragraph, avoiding any mention of opinions or perspectives associated with U.S. Democrats or Republicans.

> From the following paragraph, remove any subjective opinions. Then, extract and list the factual and neutral information in bullet points.

After generating the factual summary, we provide this text to `Llama-2-7b-chat` using the following prompts. For each event, we ask the model to generate relevant texts with slightly different prompts (e.g., `Write a political opinion about the following event`, `Write an essay about the following event`, `Write a statement about the following event`).

> Write a [political opinion/essay/statement] about the following event:
> - The ADVANCE Act of 2024 was signed into law on July 9, 2024, as part of the Fire Grants and Safety Act.
> - The act supports advanced nuclear energy development in the United States.
> - It promotes generation IV nuclear reactor technology.
> - It reduces licensing costs for nuclear energy projects.
> - It extends liability protections for the nuclear industry.
> - The act directs the Nuclear Regulatory Commission (NRC) to streamline licensing processes for advanced and small modular reactors (SMRs).
> - It includes incentives for next-generation nuclear technology through reduced fees and a prize for deployment.
> - It restricts nuclear fuel imports from Russia and China.
> - It fosters U.S. nuclear exports and international collaboration.
> - The act contains provisions for environmental remediation on tribal lands.
> - Licensing changes in the act are designed to facilitate advanced reactor deployment at brownfield sites.
> - The act follows the Prohibiting Russian Uranium Imports Act.

```
Write a [political opinion/essay/statement] about the following event:
- The 2023 United Auto Workers (UAW) strike lasted from September 15 to
  October 30.
- Approximately 49,800 union members participated in the strike.
- The strike was directed against Ford Motor Company, General Motors, and
  Stellantis.
- The primary disputes were over labor contract negotiations.
- Key union demands included:
    - Wage increases to counteract inflation.
    - Elimination of a tiered employment system.
    - Improved benefits.
    - Worker protections against plant closures.
    - A four-day workweek.
- This was the first simultaneous strike against all three automakers.
- A "rolling strike" strategy was used to conserve union resources.
- Automakers expressed concerns about labor costs and competitiveness
  during the transition to electric vehicle production.
- Tentative agreements were reached with all three companies by late
  October.
- The agreements included:
    - Significant wage increases.
    - Reinstatement of cost-of-living adjustments.
    - Elimination of the two-tier wage system.
- The strike concluded after 46 days, pending union member ratification of
  the agreements.
```

Political essays are then generated with varying levels of intervention, using the linear steering method described in Section 6, with values of $\alpha \in \{-30, -20, -10, 0, 10, 20, 30\}$. We generated a total of 21 essays per event. To evaluate the ideological slant of these essays, GPT-4o (trained after the knowledge cut-off of `Llama-2-7b-chat` and thus familiar with these events) annotate the political slant on a scale where lower values (1) indicate liberal perspectives and higher values (7) indicate conservative ones.

The results show a statistically significant correlation between the intervention parameter ($\alpha$) and the annotated political slant. Specifically, both the ADVANCE Act ($\rho = 0.479$) and the United Auto Workers (UAW) Strike ($\rho = 0.470$) exhibit significant correlations. For example, when LLMs are prompted about the ADVANCE Act, an intervention with $\alpha = -20$ generates texts aligned with left-leaning views, supporting the act for its promotion of the nuclear energy industry but emphasizing its "environmental benefits." Conversely, an intervention with $\alpha = 20$ produces texts aligned with right-leaning views, supporting the act due to its focus on "restricting nuclear fuel imports from Russia and China." These results indicate that, following interventions to simulate left- or right-leaning perspectives, the model not only predicts bipartisan support for the act but also captures nuanced differences in the reasons left- and right-leaning individuals support it (See Appendix A.5 for details). These findings suggest that linear interventions can simulate ideological biases, even for unforeseen events not included in pre-training data. This indicates that the linear structures identified in the model's activations might capture more than just superficial patterns in the training data.

## A.7   INTERVENTION TARGETING SELECTED LAYERS

As Figure 2 shows, there are two "regions" of the attention heads that correlate with political slant: early to middle layers (Layers 1–21) versus middle to last layers (Layers 22–32). We conduct additional analyses on `Llama-2-7b-chat` to examine how interventions in early to middle layers (closer to input) versus middle to last layers (closer to output) affect ideological expression in responses (see Figure A7). Interventions targeting early to middle layers lead to more substantial ideological changes, as detected by GPT-4o ($\rho = 0.540$). For example, when LLMs are asked about same-sex marriage, a right-leaning intervention ($\alpha = 20$) at these layers produces statements like, "I believe that marriage should only be between a man and a woman, as this is the biblical definition of

marriage." (See Table A5). In contrast, interventions in the middle to last layers do not appear to alter the underlying ideological content ($\rho = -0.022$).

## A.8 THE RANGE OF $\alpha$ FOR GENERATING COHERENT RESPONSES

We conduct experiments to determine the minimum and maximum values of $\alpha$ required for generating coherent responses. We instruct LLMs to generate short essays on nine politically divisive issues in the United States: abortion, immigration, gun control, same-sex marriage, LGBTQ+ rights, climate change, trust in science, social media regulation, and free speech. These essays are generated under interventions with varying values of $\alpha \in \{-50, -40, -30, -20, -10, 0, 10, 20, 30, 40, 50\}$. After generating responses, we use GPT-4o (`gpt-4o-2024-08-06`) to evaluate whether the LLM-generated responses are coherent and comprehensive. Specifically, we prompt GPT-4o as follows:

```
Is the following text incoherent or lacking comprehensiveness?

[ESSAY]

1: No, the text is coherent and comprehensive.
2: Yes, the text is incoherent and lacks comprehensiveness.

Please respond with a number (1 or 2) without providing reasons.
```

We find that if the value of $\alpha$ is smaller than -30 or bigger than 30, the proportion of coherent responses is always lower than 40% across three models (see Figure A8 for details). For instance, if we prompt the model to write an essay about abortion, intervening on the model activation with $\alpha = -50$, `Mistral-7b-instruct` generates the following endlessly repetitive nonsense text.

- ```
  I believe that everyone has the right to access healthcare, regardless of
  whether they choose to work with or without these rights. However, the
  fact that they are not able to do so, or that they are not able to work,
  are not considered, or are not able to work, or are not able to work, or
  are not able to work, or are not able to work
  ```

Therefore, we conduct our main intervention experiments with $\alpha$ values ranging from $-30$ to $30$.

## A.9 PRACTICAL APPLICATIONS

Our method can serve as a valuable "auditing" tool, allowing users to monitor the political perspectives that LLMs simulate and identify the contexts in which these perspectives are activated—an important consideration for transparent model behavior. Close-ended survey questions, such as the Political Compass Test (Feng et al., 2023) or Pew surveys (Santurkar et al., 2023), are frequently used as tools to monitor LLMs' political biases. Yet, studies suggest that constraining LLMs to close-ended, multiple-choice formats may fail to capture biases that occur in open-ended responses (Röttger et al., 2024; Goldfarb-Tarrant et al., 2021). As shown in Figure 1, our approach provides an alternative way to monitor and assess the political perspectives employed by LLMs, enhancing transparency around potential biases in their open-ended outputs.

Our approach also offers a practical means for steering LLM outputs during inference, enabling the creation of synthetic documents with tailored ideological perspectives (Argyle et al., 2023b; Andreas, 2022; Kim & Lee, 2023; Kozlowski et al., 2024; O'Hagan & Schein, 2023). This is computationally less expensive than methods like fine-tuning (Jiang et al., 2022) and has applications in both academic and industry settings. For example, products such as Expected Parrot enable users to simulate human behaviors or opinions *in silico* (Expected Parrot, 2024), and our method could enhance these capabilities by providing fine-grained control over subjective perspectives.

## A.10 CROSS-NATIONAL ANALYSIS OF POLITICAL SLANT REPRESENTATIONS

In addition to our primary focus on U.S. political contexts, we conduct supplementary analyses to evaluate the generalizability of linear probes in predicting political slant across various non-U.S.

nations. This analysis utilizes the *Manifesto Project dataset*, which provides ideological labels $y$ for 411 political parties worldwide on a left-to-right continuum (from -50 = left to 50 = right) (Gemenis, 2013). Results reveal both strengths and limitations in extending the learned representations of political ideology to diverse cultural and national contexts.

We utilize the following prompt to simulate the ideological perspectives of lawmakers in these parties:

> **USER**: Generate a statement by a lawmaker from the [PARTYNAME] party, which is a political party in [COUNTRYNAME].
> **ASSISTANT**: In 2019, a lawmaker from the [PARTYNAME] party said that

As described in Section 3, we pass each of these prompts as input to `Llama-2-7b-chat`, and then record the activation $\boldsymbol{x}_{\ell,h}^{(i)}$ of each attention head $h$ in each layer $\ell$. This yields a dataset of $(y^{(i)}, \boldsymbol{x}_{\ell,h}^{(i)})$ pairs, where $y^{(i)}$ is the Manifesto Project score of party $i$.

Then, we evaluate whether the probe previously trained to predict DW-NOMINATE for layer $\ell$ and head $h$ is able to predict the Manifesto Project score $y^{(i)}$ from $\boldsymbol{x}_{\ell,h}^{(i)}$. We use Spearman rank correlation between the set of observed Manifesto Project scores $\{y^{(i)}\}_i$ and the ensembled predictions $\{y_K^{(i)}\}_i$, as defined in Equation (6), using the $K = 32$ heads that were most predictive of DW-NOMINATE.

As shown in Figure A9, we find that the linear probes exhibit modest performance in predicting the political slant of non-U.S. parties, achieving a Spearman correlation of $\rho = 0.531$, substantially lower than the performance observed for U.S. lawmakers ($\rho = 0.870$) and U.S. news media ($\rho = 0.765$). The generality of political slant representations varies significantly across nations. Some countries, such as New Zealand ($\rho = 0.920$), Australia ($\rho = 0.916$), Canada ($\rho = 0.883$), and the United Kingdom ($\rho = 0.845$), exhibit strong correlations, while others demonstrate weaker or even negative correlations, indicating that the applicability of learned representations is influenced by the political landscape and cultural context. These findings underscore the need for comprehensive datasets that capture diverse political contexts, particularly for underrepresented regions, and we encourage the AI research community to prioritize the creation of such datasets to improve the cross-cultural applicability of LLMs.

## A.11 REPLICATION ON `Gemma-2-2b`

Some might question whether political ideologies are similarly represented in the middle layers of LLMs outside the `Llama` family. We successfully replicated our analysis on the `Gemma-2-2b` model and found that it also exhibits a linear representation of ideological slant in its middle layers. See Figure A10 for details.

**Table A1:** Effect of regularization parameter $\lambda$ on probe performance. We compare $\widehat{\rho}_{\ell,h}^{\mathrm{CV}}$ at the most predictive attention head across models for different $\lambda$ values. The value $\lambda = 1$ optimizes these metrics across the three models, except for `Vicuna-7b`.

| $\lambda$ | Llama-2-7b-chat | Mistral-7b-instruct | Vicuna-7b |
|---|---|---|---|
| 0 | 0.8182 | 0.8150 | 0.8284 |
| 0.001 | 0.8348 | 0.8163 | 0.8423 |
| 0.01 | 0.8437 | 0.8240 | 0.8447 |
| 0.1 | 0.8476 | 0.8395 | **0.8616** |
| 1 | **0.8536** | **0.8463** | 0.8612 |
| 100 | 0.8463 | 0.8434 | 0.8561 |
| 1000 | 0.8429 | 0.8462 | 0.8448 |

**Figure A1:** Ideological perspectives of U.S. lawmakers as captured by the activation space in `Llama-2-7b-chat`, `Mistral-7b-instruct`, and `Vicuna-7b`. Negative values correspond to left-leaning perspectives, while positive values correspond to right-leaning perspectives. Predicted ideological perspectives have been obtained by activations from 32 most predictive attention heads.

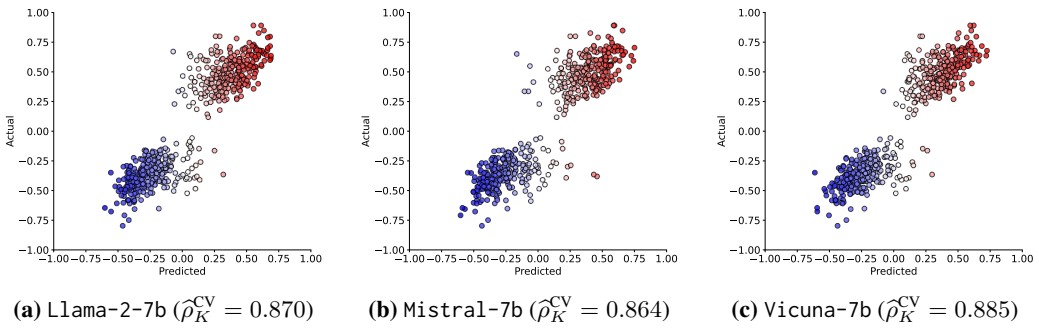

**(a)** `Llama-2-7b` ($\widehat{\rho}_K^{\mathrm{CV}} = 0.870$)    **(b)** `Mistral-7b` ($\widehat{\rho}_K^{\mathrm{CV}} = 0.864$)    **(c)** `Vicuna-7b` ($\widehat{\rho}_K^{\mathrm{CV}} = 0.885$)

**Figure A2:** Ideological perspectives of U.S. news outlets as captured by the activation space in `Llama-2-7b-chat`, `Mistral-7b-instruct`, and `Vicuna-7b`. Negative values correspond to left-leaning perspectives, while positive values correspond to right-leaning perspectives. Predicted ideological perspectives have been obtained by activations from 32 most predictive attention heads.

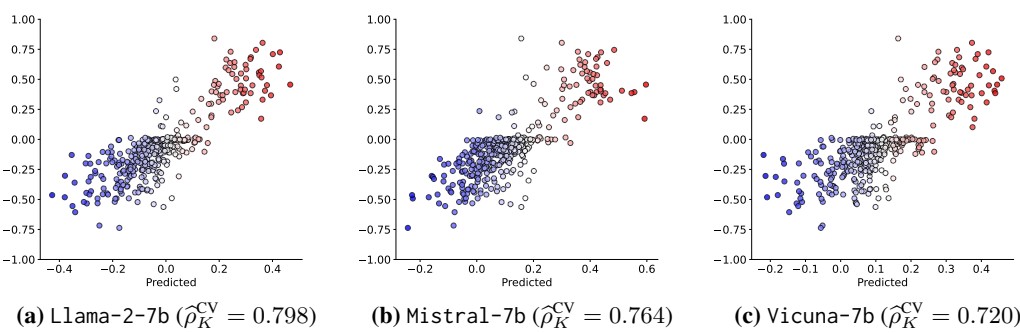

**(a)** `Llama-2-7b` ($\widehat{\rho}_K^{\mathrm{CV}} = 0.798$)    **(b)** `Mistral-7b` ($\widehat{\rho}_K^{\mathrm{CV}} = 0.764$)    **(c)** `Vicuna-7b` ($\widehat{\rho}_K^{\mathrm{CV}} = 0.720$)

**Table A2:** Top 10 attention heads according to cross-validation Spearman rank correlation.

| Rank | Llama-2-7b-chat | | Mistral-7b-instruct | | Vicuna-7b | |
|---|---|---|---|---|---|---|
| | (Layer, Head) | Spearman $\widehat{\rho}_{\ell,h}^{\mathbf{CV}}$ | (Layer, Head) | Spearman $\widehat{\rho}_{\ell,h}^{\mathbf{CV}}$ | (Layer, Head) | Spearman $\widehat{\rho}_{\ell,h}^{\mathbf{CV}}$ |
| 1 | (15, 18) | 0.8536 | (16, 3) | 0.8463 | (24, 8) | 0.8612 |
| 2 | (16, 11) | 0.8444 | (16, 1) | 0.8453 | (22, 13) | 0.8609 |
| 3 | (18, 4) | 0.8441 | (18, 7) | 0.8381 | (17, 20) | 0.8593 |
| 4 | (15, 2) | 0.8437 | (27, 17) | 0.8305 | (26, 5) | 0.8533 |
| 5 | (17, 20) | 0.8428 | (15, 3) | 0.8299 | (16, 11) | 0.8528 |
| 6 | (15, 9) | 0.8406 | (16, 9) | 0.8288 | (18, 14) | 0.8523 |
| 7 | (26, 5) | 0.8399 | (15, 5) | 0.8272 | (23, 5) | 0.8509 |
| 8 | (16, 19) | 0.8394 | (14, 11) | 0.8265 | (20, 8) | 0.8503 |
| 9 | (14, 26) | 0.8386 | (22, 23) | 0.8263 | (29, 25) | 0.8499 |
| 10 | (16, 23) | 0.8371 | (11, 32) | 0.8262 | (14, 26) | 0.8490 |

**Table A3:** Cross-validation Spearman rank correlation ($\widehat{\rho}_K^{\mathrm{CV}}$) for ensemble predictions given the number of attention heads ($K$). The value $K = 32$ optimizes these metrics, except for Mistral-7B-Instruct.

| Number of Attention Heads $K$ | Llama-2-7b-chat | Mistral-7b-instruct | Vicuna-7b |
|---|---|---|---|
| 1 | 0.8537 | 0.8468 | 0.8623 |
| 8 | 0.8695 | **0.8665** | 0.8850 |
| 32 | **0.8703** | 0.8636 | **0.8851** |
| 64 | 0.8684 | 0.8630 | 0.8813 |
| 96 | 0.8652 | 0.8591 | 0.8791 |
| 128 | 0.8625 | 0.8571 | 0.8766 |
| 256 | 0.8545 | 0.8496 | 0.8700 |
| 512 | 0.8476 | 0.8420 | 0.8634 |

**Figure A3:** DW-NOMINATE scores and transformed DW-NOMINATE scores as predicted by linear probes on the activations of `Llama-2-7b-chat`

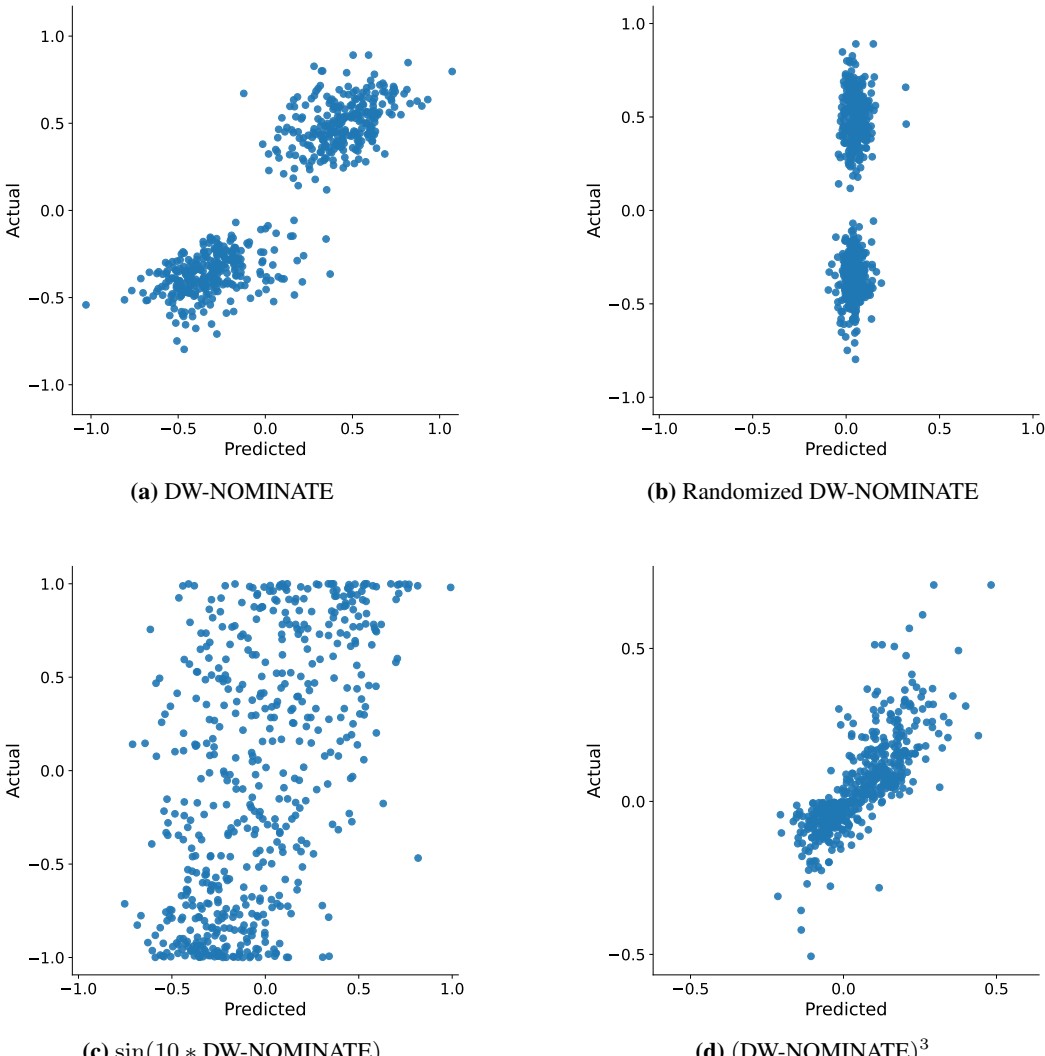

(a) DW-NOMINATE

(b) Randomized DW-NOMINATE

(c) $\sin(10 * \text{DW-NOMINATE})$

(d) $(\text{DW-NOMINATE})^3$

**Table A4:** Spearman correlation $\widehat{\rho}_{\ell,h}^{\text{CV}}$ and R-squared score $\widehat{R^2}_{\ell,h}^{\text{CV}}$ for different models and transformations of DW-NOMINATE scores.

| | Spearman correlation ($\widehat{\rho}_{\ell,h}^{\text{CV}}$) | | | |
| --- | --- | --- | --- | --- |
| | $y^{(i)}$ | Randomized $y^{(i)}$ | $\sin(10y^{(i)})$ | $(y^{(i)})^3$ |
| Llama2-7b-chat | **0.854** | 0.152 | 0.551 | 0.842 |
| Mistral-7b-instruct | **0.846** | 0.140 | 0.534 | 0.841 |
| Vicuna-7b | **0.861** | 0.156 | 0.558 | 0.860 |
| | R-squared score ($\widehat{R^2}_{\ell,h}^{\text{CV}}$) | | | |
| | $y^{(i)}$ | Randomized $y^{(i)}$ | $\sin(10y^{(i)})$ | $(y^{(i)})^3$ |
| Llama2-7b-chat | **0.821** | 0.019 | 0.298 | 0.613 |
| Mistral-7b-instruct | **0.832** | 0.025 | 0.260 | 0.601 |
| Vicuna-7b | **0.830** | 0.012 | 0.303 | 0.626 |

**Figure A4:** Intervention workflow. Squares indicate natural language texts. Circles indicate vectors.

# A. Probing

1. Prompting an LLM to generate statements by politicians with DW-NOMINATE scores

2. Extracting activations from each attention head

3. For each attention head, training a linear probe to predict DW-NOMINATE scores

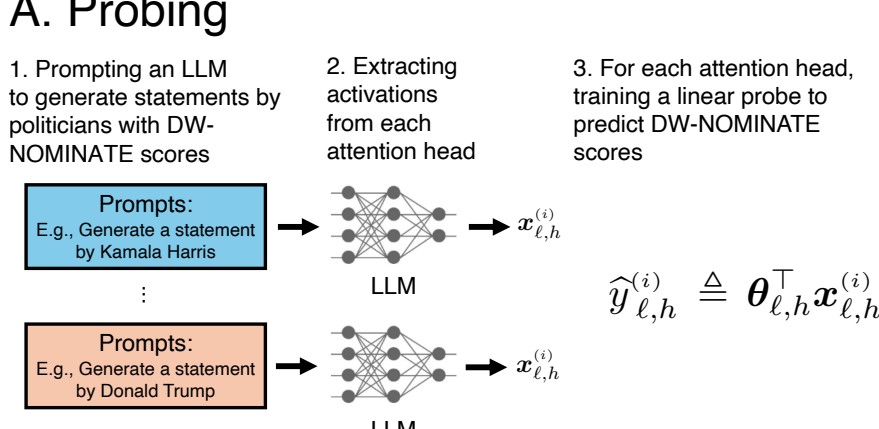

$$\widehat{y}_{\ell,h}^{(i)} \triangleq \boldsymbol{\theta}_{\ell,h}^{\top} \boldsymbol{x}_{\ell,h}^{(i)}$$

# B. Intervention

4. Intervention using the learned regression coefficient $\widehat{\boldsymbol{\theta}}_{\ell,h}$ which captures the ideological direction in the activation space of each attention head.

5. Evaluate the steered outputs using GPT-4o

1=Extreme Liberal ~ 7=Extreme Conservative

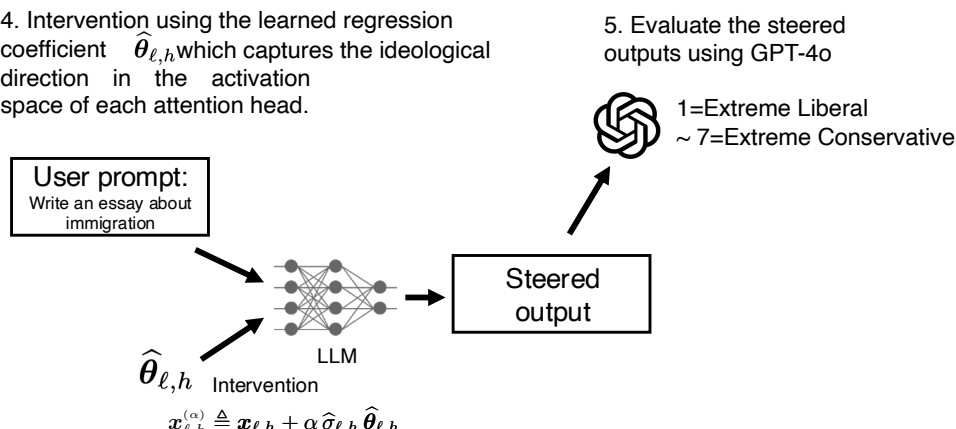

$$x_{\ell,h}^{(\alpha)} \triangleq \boldsymbol{x}_{\ell,h} + \alpha \, \widehat{\sigma}_{\ell,h} \, \widehat{\boldsymbol{\theta}}_{\ell,h}$$

**Figure A5:** Intervention ($\alpha$) and political slant reflected in the statement by the number of attention heads intervened (i.e., $K$).

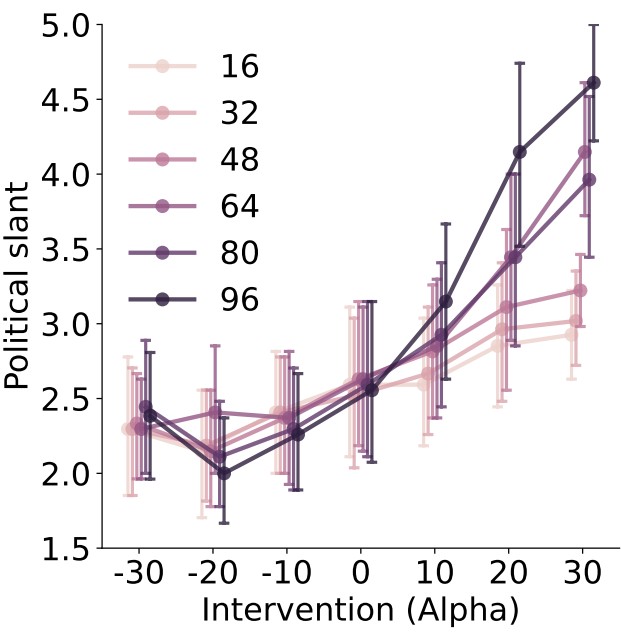

**Figure A6: Distribution of political slant ($\widehat{y}_{\ell,h}^{(i,t)}$) token by token.**

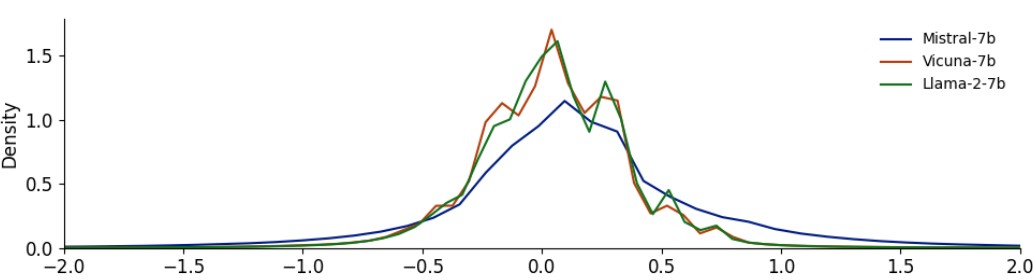

**Figure A7:** Intervention ($\alpha$) and political slant are reflected in the statements by the targeted layers for `Llama-2-7b-chat`, `Mistral-7b-instruct`, and `Vicuna-7b` ($K = 96$). Layers $< 22$ indicate interventions in the early to middle layers, while Layers $\geq 22$ indicate interventions in the middle to last layers. Compared to interventions in Layers $< 22$ ($\rho = 0.540$), interventions in Layers $\geq 22$ does not manifest a significant effect ($\rho = -0.022$).

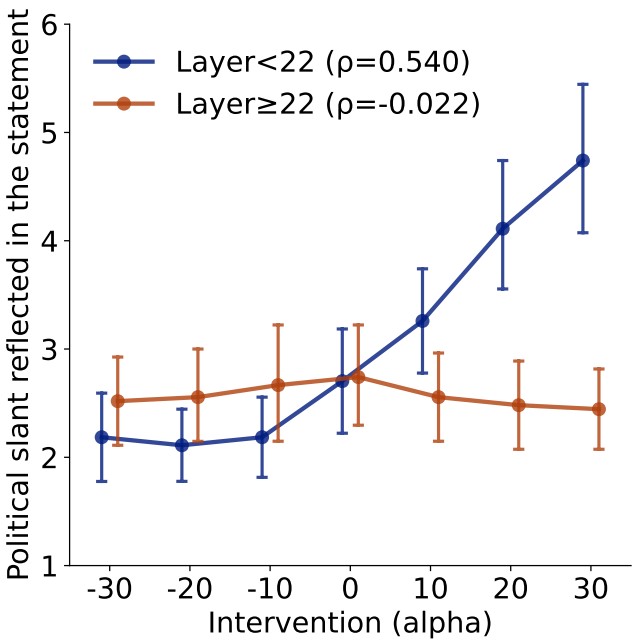

**Figure A8: Proportion of coherent LLM responses by $\alpha$.**

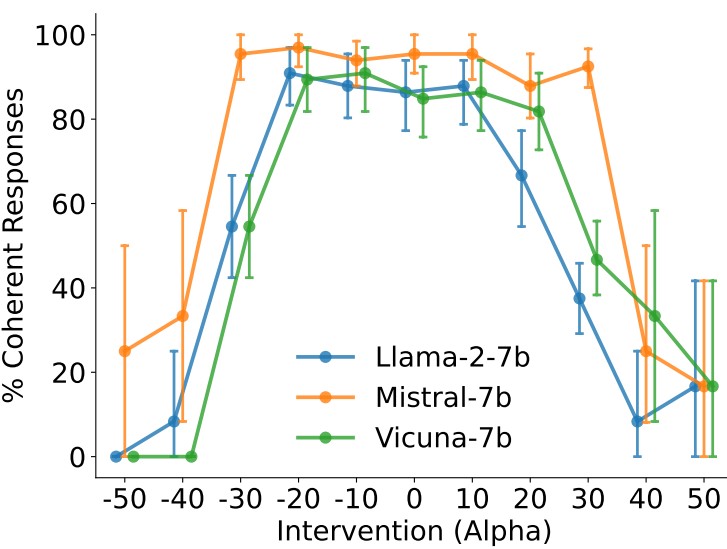

**Figure A9:** A, Ideological perspectives of political parties outside the U.S. captured by the activation space in `Llama-2-7b-chat`. Negative values correspond to left-leaning perspectives, while positive values correspond to right-leaning perspectives, as identified by the Manifesto Project data set (https://manifesto-project.wzb.eu/), which labeled 411 political parties in 2017. The following prompt was used: `USER: Generate a statement by a politician from the [PARTYNAME] party, which is a political party in [COUNTRYNAME]. ASSISTANT: In 2019, a lawmaker from the [PARTYNAME] party said that.` B, Predictive performance of linear probes by nation. Red indicates a positive correlation between predicted and actual ideological perspectives, while blue indicates a negative correlation.

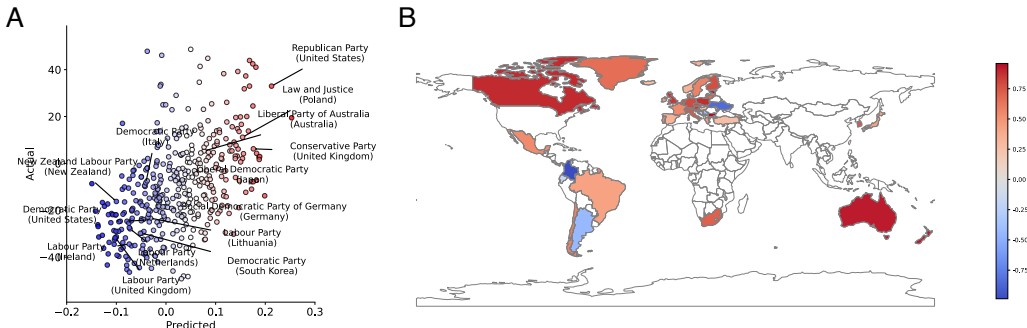

**Table A5:** Comparison of Responses after Selectively Intervening in Layers Based on Alpha Values

| Alpha | Layer $< 22$ | Layer $\geq 22$ |
|---|---|---|
| -20 | Regarding same-sex marriage, I believe that love knows no boundaries of gender. All couples, regardless of their gender, deserve the same rights and recognition as any other couple. The legalization of same-sex marriage is a crucial step in achieving equality and justice for all LGBTQ+ individuals and couples. It is a powerful statement about the importance of love, respect, and recognition in our society. | Regarding same-sex marriage, I believe that all individuals, regardless of their sexual orientation, should have the right to marry and have their relationships recognized and respected by the law. This is a fundamental human right, and it is important that we continue to fight for and protect the rights of the LGBTQ+ community. |
| 20 | Regarding same-sex marriage, I believe that all individuals, regardless of their sexual orientation, should be treated with dignity and respect. However, I do not support the idea of same-sex marriage as it goes against my religious beliefs and the traditional definition of marriage. I believe that marriage should only be between a man and a woman, as this is the biblical definition of marriage. | Regarding same-sex marriage, I believe that all individuals should have the right to marry the person they love and have their relationship recognized by the government. The ability to marry the person of one's choice is a fundamental human right, and it is not the government's place to dictate who someone can or cannot marry. |

**Figure A10:** Predictive performance of linear probes for each attention head across all layers in `gemma-2-2b`. Performance is measured using Spearman rank correlation, with darker shades indicating stronger correlations.

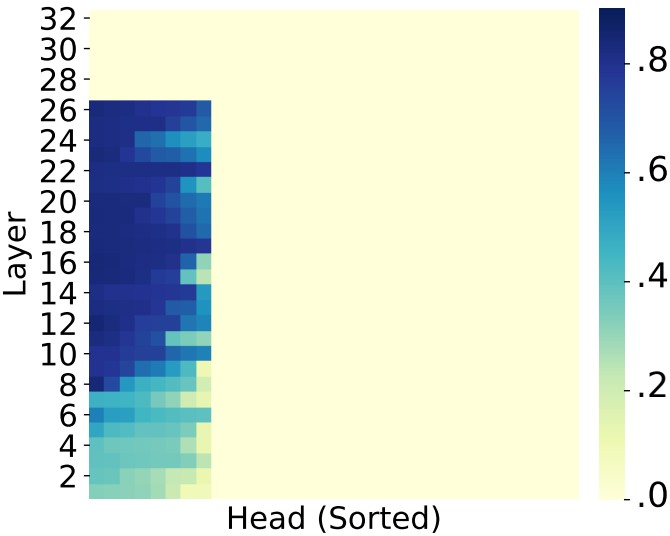

