# OpenReview forum: "Linear Representations of Political Perspective Emerge in Large Language Models"
_ICLR.cc/2025/Conference — ICLR 2025 Oral_

### Official Review · Reviewer_Huro · 2024-10-28

**Soundness:** 3
**Presentation:** 3
**Contribution:** 3
**Rating:** 6
**Confidence:** 4

**Summary:**

This paper presents an investigation into how LLMs encode political ideology, specifically focusing on U.S. politics. The authors claim that LLMs develop a linear representation of political perspectives in their activation space and demonstrate that these representations are primarily located in the middle layers of transformer models. By training a linear probe, the paper shows that political ideologies can be linearly separated within certain attention heads. The authors further demonstrate that intervening in these attention heads allows them to steer the political bias of generated text along a liberal-conservative spectrum without using explicit prompts.

**Strengths:**

- The paper is novel in its application of linear probing to the representation of political ideology, which has not been extensively explored before in the context of LLMs. This contributes to the growing body of research on model interpretability and bias detection.
- The experiments are well-executed, and the use of multiple open-source models (Llama-2-7b, Mistral-7b, Vicuna-7b) to ensure reproducibility is commendable. The intervention analysis demonstrates an interesting way to manipulate model behavior based on internal activations.
- The paper tackles a highly relevant issue, as understanding and controlling political bias in LLMs is a critical challenge, especially given the role AI systems play in information dissemination and public discourse.

**Weaknesses:**

- The key contribution of the paper—demonstrating linear separability of political ideologies—is not as novel or profound as the authors suggest. Linear separability is an expected feature of transformer models, particularly in middle layers where complex features are encoded. The use of a linear probe to classify two categories (liberal vs. conservative) is a relatively simple task that doesn’t fully substantiate the broader claim of a "linear representation hypothesis" in LLMs. The results merely demonstrate linear separability rather than showing that political ideology is fundamentally encoded as a linear feature across multiple dimensions, which would be a much more significant finding.
- The paper seems to conflate linear separability with the broader notion of linear representation as discussed in interpretability literature. The linear representation hypothesis posits that various features, such as gender or sentiment, can be combined and manipulated linearly within a model’s activation space. Simply demonstrating that political ideology can be classified with a linear probe doesn’t fully support this hypothesis. The authors don’t explore whether political ideology can be combined with other features in a meaningful, linear fashion, nor do they show how this linearity generalizes to other tasks or representations beyond political ideology.
- While the paper uses interventions to modify the political bias of generated text, it doesn’t convincingly demonstrate the causal relationship between specific model activations and ideological outputs. It remains unclear whether the identified linear structures are genuinely capturing political bias or merely reflecting superficial patterns in the training data. The interventions themselves seem to rely on surface-level associations rather than a deeper understanding of how political ideologies are encoded and combined within the model.

**Questions:**

- Simulating Ideological Perspectives (Line 185): When you state that "LLMs must simulate the subjective, ideological perspectives of the given politicians or news media," how do you ensure that the model consistently selects an accurate ideological stance, rather than a random or mixed perspective, especially if it lacks full factual context? Could you provide additional details on how factual alignment with specific perspectives is verified in the model’s responses?
- Reference to Political Slant as a ‘Linear’ Spectrum (Line 133): The paper mentions that "social scientists have shown that humans conceptualize political slant as a ‘linear’ spectrum." Could you provide specific references to support this claim? Additionally, how does this linear conceptualization integrate with political science literature, which often models ideology as multi-dimensional? Exploring this could clarify how your approach fits within or challenges existing theories of political ideology.

---

> ### Author Response · Authors · 2024-11-21
> **Official Comment by Authors (Part 1/3)**
>
> We thank Reviewer Huro for their thoughtful engagement and valuable feedback on our manuscript. We have made substantial revisions and **submitted an updated version of the manuscript** in accord with these expressed concerns and constructive suggestions.
>
> > **Weakness 3: While the paper uses interventions to modify the political bias of generated text, it doesn’t convincingly demonstrate the causal relationship between specific model activations and ideological outputs. It remains unclear whether the identified linear structures are genuinely capturing political bias or merely reflecting superficial patterns in the training data. The interventions themselves seem to rely on surface-level associations rather than a deeper understanding of how political ideologies are encoded and combined within the model.**
>
> We appreciate the reviewer raising the concern that the identified linear structures in the model might reflect “superficial patterns in the training data” rather than genuinely capturing political bias. To address this, we conducted additional analyses showing that the linear structure can modulate ideological outputs even when the model generates opinions about events that occurred after its knowledge cut-off (i.e., not in the pre-training data). Specifically, we examined the model’s responses to two politically significant events that occurred after the knowledge cut-off date of September 2022 for Llama-2-7b-chat’s pre-training data. These events are:
>
>   - **Accelerating Deployment of Versatile, Advanced Nuclear for Clean Energy (ADVANCE) Act (March 2023)**
>   - **United Auto Workers (UAW) Strike (September 2023)**
>
> First, we verified that the model lacked prior knowledge of the event by prompting it with the questions, “Do you have information about the ADVANCE Act in 2023? Respond in Yes or No.” and “Do you have information about the United Auto Workers (UAW) Strike war in 2023? Respond in Yes or No.” The model responded that they do not know the events, showing that any generated opinions were not influenced by patterns present in the training data.
>
> Next, we provided a factual summary of each event (see Appendix A.6 for the prompt) and asked the model to write political essays about the event with varying levels of ideological intervention, using the linear steering method described in Section 3.2. Essays were generated across intervention values $\alpha$ (-30, -20, -10, 0, 10, 20, 30) with varying hyperparameters $K$ (64, 80, 96), resulting in 21 essays per event. GPT-4, which was trained after the knowledge cut-off of Llama-2-7b-chat and knew about these events, was employed to annotate the political slant of the essays.
>
> **The results showed a statistically significant correlation between the intervention parameter ($\alpha$) and the annotated political slant.** Specifically, both the ADVANCE Act ($r=0.479$, $p=0.028$, $N=21$) and the UAW Strike ($r=0.470$, $p=0.031$, $N=21$) exhibited significant correlations. For example, when prompted to write political essays about the ADVANCE Act, an intervention with $\alpha=-20$ generated texts aligned with left-leaning views, supporting the act for its promotion of nuclear energy industries but emphasizing its "environmental benefits." Conversely, an intervention with $\alpha=20$ produced texts aligned with right-leaning views, supporting the act due to its focus on "restricting nuclear fuel imports from other countries." These results indicate that, following interventions to simulate left- or right-leaning perspectives, the model predicts not only bipartisan support for the act but also captures nuanced differences in the reasons underlying left-leaning and right-leaning individuals' support for it. Please refer to Table A4 for details.
>
> These findings indicate that the linear structures identified in the model’s activation space capture more than surface-level patterns from training data; they reflect latent ideological representations that can be dynamically adjusted. The generated essays demonstrated differences in structure, framing, and argumentation—beyond simple lexical differences such as “welfare” versus “entitlements”—providing evidence of non-trivial simulation of ideological bias.

---

> ### Author Response · Authors · 2024-11-21
> **Official Comment by Authors (Part 2/3)**
>
> > **Question 1: Simulating Ideological Perspectives (Line 185): When you state that "LLMs must simulate the subjective, ideological perspectives of the given politicians or news media," how do you ensure that the model consistently selects an accurate ideological stance, rather than a random or mixed perspective, especially if it lacks full factual context? Could you provide additional details on how factual alignment with specific perspectives is verified in the model’s responses?**
>
>
> As discussed above, we found that the model consistently generated ideologically coherent perspectives by intervening in its activation space to simulate subjective opinions about the ADVANCE Act or the UAW Strike. This case study demonstrates the model's ability to produce ideologically consistent outputs even in the absence of factual context about political reactions in the training data.
>
> That said, we acknowledge that Line 185 in the original draft overstated the claim that LLMs “must” simulate ideological perspectives. In the revised manuscript, we have softened this language to avoid overgeneralization.
>
> > **Question 2: Reference to Political Slant as a ‘Linear’ Spectrum (Line 133): The paper mentions that "social scientists have shown that humans conceptualize political slant as a ‘linear’ spectrum." Could you provide specific references to support this claim? Additionally, how does this linear conceptualization integrate with political science literature, which often models ideology as multi-dimensional? Exploring this could clarify how your approach fits within or challenges existing theories of political ideology.**
>
> Our conceptualization of political slant as a linear spectrum is informed by the theory of “partisan sorting,” which is well-documented (Baldassarri and Gelman, 2008; Fiorina and Abrams, 2008; Levendusky, 2009; Noel, 2014; Layman et al., 2006). Partisan sorting suggests that U.S. political identity is increasingly aligned along a single left-right continuum (e.g., Left = Strong Democrat, Right = Strong Republican), with heightened ideological consistency within each political affiliation. This unidimensional model is supported by empirical research showing that partisan alignment correlates with a broad range of issue stances, including economic policies, social issues like abortion and morality, and environmental concerns (Baldassarri and Gelman, 2008; Fiorina and Abrams, 2008). Polarization among political elites has intensified this sorting process in recent decades, fostering a more unidimensional political landscape than in prior decades (Baldassarri and Gelman, 2008).
>
> While political ideology can be multi-dimensional, studies indicate that in practice, U.S. political discourse is dominated by a left-right dimension, which simplifies and aligns otherwise diverse issue stances along a single axis (Baldassarri and Gelman, 2008; Fiorina and Abrams, 2008; Levendusky, 2009; Noel, 2014; Layman et al., 2006). Moreover, the left-right continuum is not only prevalent in political science research but also widely adopted in mainstream and social media as a heuristic for discussing political biases (Bruin et al., 2023; Waller and Anderson, 2021). By visualizing LLM outputs along this continuum (e.g., Figure 1), we enhance the interpretability of LLM biases, making them more accessible to both researchers and general users.
>
> Please refer to Section 2: Linear Representations of Political Ideology in the revised draft.

---

> ### Author Response · Authors · 2024-11-21
> **Official Comment by Authors (Part 3/3)**
>
> > **Weakness 1, 2: The key contribution of the paper—demonstrating linear separability of political ideologies—is not as novel or profound as the authors suggest. Linear separability is an expected feature of transformer models, particularly in middle layers where complex features are encoded. The use of a linear probe to classify two categories (liberal vs. conservative) is a relatively simple task that doesn’t fully substantiate the broader claim of a "linear representation hypothesis" in LLMs. The results merely demonstrate linear separability rather than showing that political ideology is fundamentally encoded as a linear feature across multiple dimensions, which would be a much more significant finding.**
>
> > **The paper seems to conflate linear separability with the broader notion of linear representation as discussed in interpretability literature. The linear representation hypothesis posits that various features, such as gender or sentiment, can be combined and manipulated linearly within a model’s activation space. Simply demonstrating that political ideology can be classified with a linear probe doesn’t fully support this hypothesis. The authors don’t explore whether political ideology can be combined with other features in a meaningful, linear fashion, nor do they show how this linearity generalizes to other tasks or representations beyond political ideology.**
>
> We regret that the conceptualization of linear representations informing our work was not sufficiently articulated or emphasized in the original draft. In the revised version, we clarify that our study builds on a well-established line of research exploring the linearity of features and concepts in neural networks. Specifically, we adopt the following definition of “linear representation”:
>
> “Features within neural networks are represented linearly, meaning the presence or strength of a feature can be determined by projecting the relevant activation onto a feature vector” (Mikolov et al., 2013; Olah et al., 2020; Elhage et al., 2022; Gurnee and Tegmark, 2024).
>
> We would also like to clarify that the "left-right" or "liberal-conservative" dimension is not a binary category but a continuous feature. Our ground-truth measures of political ideology, such as DW-NOMINATE and Ad Fontes Media, are also continuous features (e.g., distinguishing between strong and slightly liberal positions). While much of the research on the linear representation hypothesis has focused on binary or categorical features, Gurnee and Tegmark (2023) suggest that the linear representation of continuous features, such as space and time, is less investigated. Our research contributes to this body of evidence by demonstrating that continuous features, like political ideology, are also linearly represented.
>
> In addition, we have expanded the “Conclusion and Limitations” section to propose future research directions, such as investigating whether political ideology can be combined with other features in a meaningful linear fashion or exploring how this linearity generalizes to tasks and representations beyond political ideology.
>
> We also note a growing interest in the machine learning community regarding the political biases of AI (e.g., Santurkar et al., ICML 2023). This aligns with ICLR 2025’s focus on “societal considerations of representation learning, including fairness, safety, privacy, interpretability, and explainability” (see https://iclr.cc/). Furthermore, the relevance of political bias in AI has been highlighted by recent studies and events, such as the U.S. presidential election (e.g., see Potter et al., 2024), underscoring the timeliness and societal significance of our research. We note that, to the best of our knowledge, this research is the first to examine the representation of political biases (as highlighted by Reviewer vb63 in Weakness 1), and we believe our work makes a meaningful contribution to this critical area of study.

---

> ### Comment · Reviewer_Huro · 2024-11-25
>
> I thank the authors for their detailed response to all reviewers and the significant revisions to the manuscript. The clarifications on linear representations, political ideology, and other additions strengthen the paper’s contributions and address my key concerns.
>
> I have updated my score as follows:
>
>
> **Contribution**: 2 → 3
> **Rating**: 5 → 6

---

### Official Review · Reviewer_T21J · 2024-11-01

**Soundness:** 4
**Presentation:** 3
**Contribution:** 3
**Rating:** 8
**Confidence:** 4

**Summary:**

This study shows how LLMs, like Llama-2-7b-chat, Mistral-7b, and Vicuna-7b, represent and simulate ideological perspectives in their hidden layers. Researchers probed these layers and identified a specific set of attention heads that represent particular political views, uncovering patterns they call the "geometry of perspective." They claim that these attention head activations linearly align with LLM viewpoints on a liberal-to-conservative scale, correlating with known political ideologies. They also found that adjusting these attention heads can steer model responses without extra prompts, shaping the output toward liberal, center, or conservative viewpoints. I feel this work could allow closer monitoring and control of ideological perspectives in LLM responses, which could be a powerful tool for transparency and balance in AI-generated content.

**Strengths:**

1. The overall problem statement and motivation are very interesting and could be useful in real-world applications.
2. They used a simple yet effective approach by examining attention heads with established probing methods, applying a regression model on network activations to predict annotated labels of input or output data.
3. After probing, they attempted intervention with the model, trying to steer outputs to gain more insights—which was very interesting.

**Weaknesses:**

1. I think more details should be added in Section 3.2, *Intervention*. It is very unclear.
   - Comments:
      - In line 282, $\theta_{l,h}$ is described as a steering vector, but in line 228, it is referred to as attention coefficients. Are they the same? If not, why are the notations the same?
      - If $\theta_{l,h}$ represents attention coefficients from the Ridge regression model, how can it be applied directly in the attention mechanism? Are the shapes of both vectors/tensors the same? Also, as they originate from different areas, how are they connected?
      - If $\theta_{l,h}$ is not the attention coefficient in line 282, how is it calculated? How do we maintain its shape or connect it with other components?
      - In line 288, the definition, purpose, and use of $\sigma_{l,h}$ are also unclear.
      - In line 291, it states, *We target the K most predictive attention heads for the intervention.* What is the value of $K$ here? Is it selected from the coefficients of the Ridge regression?
   - Asks:
      - Clearly define $\theta_{l,h}$ and explain how it relates to both the ridge regression model and the intervention process.
      - Explain the process of applying the ridge regression coefficients to the attention mechanism, including any necessary transformations.
      - Provide a clear definition and explanation for $\sigma_{l,h}$.
      - Specify how K is determined and its relationship to the ridge regression results.

2. In Section 3.6, GPT-4 as an evaluator is experiment-friendly, but isn’t there a chance we might reinforce hidden biases of LLMs? For instance, if the LLM-based evaluator has a bias toward the right (assuming right-leaning perspectives are correct), could it potentially score right-leaning statements more favorably, as this bias would seem normal to the evaluator model? Usually, some human annotated experiments are expected to prove that the LLM evaluator is replicating actual human/proper responses.
    - Asks:
      - Acknowledge and discuss the potential for bias when using an LLM as an evaluator.
      - Do some experiments to validate GPT-4's evaluations against human annotations, even if on a smaller scale.
      - Discuss how these factors might mitigate or account for potential biases in the evaluation process.

3. A discussion could be added to connect the sections *Simulating Subjective Perspectives using LLMs* and *Political Bias of LLMs* in related works, discussing the potential societal impact of this study. Currently, the findings feel somewhat disconnected from real-world applications. A brief discussion section could help.
    - Asks:
      - Add a brief discussion section that explicitly connects your findings to potential real-world applications.
      - Explore how your method for identifying and intervening in political bias could be applied in practical scenarios.
      - Discuss the ethical implications and potential societal impacts of your findings.

4. Although limitations mention that a U.S.-centric focus may reduce relevance in global contexts, the topic of political discourse and presence of strongly opposing views differs in many countries. As a result, the scope is limited; also, it’s uncertain if LLMs will adapt as accurately to culturally distinct perspectives (such as those in South Asia) due to data limitations.

5. Writing quality needs improvement, with several missing details. I’ve added these in the *Questions** below.

6. From a theoretical standpoint, the methods used are derived from previous works. While the study provides an interesting new application, I still feel this work lacks novel theoretical contributions.

**Questions:**

1. Check weakness 1.
2. In line 223, it says, "*We train a ridge regression model using U.S. politician data.*" Does *U.S. politician data* refer to the extracted activations? Is there any train-test split, or how does this part work?
3. In line 344, was *2-fold cross-validation* done only on the Ridge regression model, or did you also collect attention heads twice?
4. Can you provide more details on the x-axis legends (values of each line) in Figure 3?
5. Clarify the terms *accuracy* and *Spearman rank correlation* in Section 4.1. Why is each one used?
6. How is Figure 4 generated? Is it based on the DW-NOMINATE scores from the datasets and attention, or does it use the attention and the political aspect of the prompt?
7. In line 417, it says, *the strongest correlation between intervention and political slant*, and the previous line mentions achieving a higher degree of conservative political slant. Is the correlation value based only on steering toward a conservative slant, or on all three (liberal, conservative, and neutral)? Again, how are these values measured, and with which inputs, outputs, or parameters?
8. What are the hyperparameters of Ridge regression model? Is it tuned? If yes, how? Is it tuned for all different situations separately, or only one model is trained - how does this work? If it is not tuned, why?
9. Adding a figure showing different parts of how the model works from start to finish, like a pipeline, could help the audience better understand the approach.

---

> ### Author Response · Authors · 2024-11-21
> **Official Comment by Authors (Part 1/6)**
>
> We thank Reviewer T21J for their thoughtful engagement and valuable feedback on our manuscript. We have made substantial revisions and **submitted an updated version of the manuscript** in accord with these expressed concerns and constructive suggestions.
>
> > **Weakness 1. I think more details should be added in Section 3.2, Intervention. It is very unclear.**
>
> Please refer to our responses to Question 1 below.
>
> > **Weakness 2. In Section 3.6, GPT-4 as an evaluator is experiment-friendly, but isn’t there a chance we might reinforce hidden biases of LLMs? For instance, if the LLM-based evaluator has a bias toward the right (assuming right-leaning perspectives are correct), could it potentially score right-leaning statements more favorably, as this bias would seem normal to the evaluator model? Usually, some human annotated experiments are expected to prove that the LLM evaluator is replicating actual human/proper responses.**
>
> We thought this was an important point and ran a survey-based validation, which is now reported in the revised draft (Appendix A.2.3). We have now validated GPT-4’s evaluations against politically balanced human annotators, and the results demonstrate strong alignment between GPT-4’s ratings and human annotations.
>
> Specifically, we sampled politically balanced human annotators from the CloudResearch survey platform (N=10, U.S. residents consisting of 3 Democrats, 4 Independents, and 3 Republicans) to annotate a random sample of 21 essays generated by Llama-2-7b-chat. After averaging the scores provided by these human annotators, we measured inter-rater reliability between GPT-4 and the human annotators’ average scores. We found a very high inter-rater reliability ($ICC(A,1) = .91$), supporting the validity of GPT-4 in annotating political slant. Additionally, the Spearman correlation between GPT-4 and the average human scores was very high ($r = 0.952, p < 10^{-10}$). Our findings are consistent with O'Hagan and Schein (2023)’s work, which demonstrated that LLMs can reliably reflect established measures of ideological slant.
>
> We have also expanded the discussion to emphasize the importance of addressing potential biases in LLM evaluations. We recommend that future research using our methods continue to validate LLM-generated annotations against human annotations to mitigate any inherent biases. See “Conclusion and Limitations” section for details.
>
> > **Weakness 3: A discussion could be added to connect the sections Simulating Subjective Perspectives using LLMs and Political Bias of LLMs in related works, discussing the potential societal impact of this study. Currently, the findings feel somewhat disconnected from real-world applications. A brief discussion section could help.**
>
> We have now added a discussion section that explicitly addresses these points.
>
>   - **Practical Applications (see Appendix A.8):**
>     - Our method can serve as a valuable “auditing” tool, allowing users to monitor the political perspectives that LLMs simulate and identify the contexts in which these perspectives are activated—an important consideration for transparent model behavior. While close-ended survey questions, such as those in the Political Compass Test (Feng et al., 2023) or Pew surveys (Santurkar et al., 2023), are commonly used to assess LLMs’ political biases, this approach may overlook biases present in open-ended responses (Röttger et al., 2024; Goldfarb-Tarrant et al., 2021). As shown in Figure 1, our approach provides an alternative way to monitor and assess the political perspectives employed by LLMs, enhancing transparency around potential biases in their open-ended outputs.
>     - Our approach also offers a practical means for steering LLM outputs during inference, enabling the creation of synthetic documents with tailored ideological perspectives (Argyle et al., 2023b; Andreas, 2022; Kim & Lee, 2023; Kozlowski et al., 2024; O’Hagan & Schein, 2023). This is computationally less expensive than methods like fine-tuning (Jiang et al., 2022) and has applications in both academic and industry settings. For example, products such as Expected Parrot enable users to simulate human behaviors or opinions in silico (https://www.expectedparrot.com/), and our method can enhance these capabilities by providing fine-grained control over political perspectives.

---

> ### Author Response · Authors · 2024-11-21
> **Official Comment by Authors (Part 2/6)**
>
> - **Practical Scenario (see Appendix A.6):** We included a case study focusing on practical scenarios where users ask LLMs about politically relevant events not covered in the models’ training data. The practical application of LLMs in debating ongoing political matters has been studied in recent literature (e.g., Costello et al., 2024). Specifically, we prompted LLMs to generate essays about the *Accelerating Deployment of Versatile, Advanced Nuclear for Clean Energy (ADVANCE) Act* (March 2023) and the *UAW Strike* (September 2023). Since these events occurred after the models’ knowledge cutoff, our case study demonstrates how ideological steering can simulate perspectives on emerging issues that LLMs have not explicitly learned about. After the steering, the model successfully generated ideological texts regarding these events, suggesting both potential applications of using the technique to generate ideological texts for debates (e.g., Costello et al., 2024) and social science research (e.g., Argyle et al., 2023). Simultaneously, it suggests potential risks of misusing the technique to generate biased texts.
>
> - **Ethical implications and potential societal impacts (see Ethics Statement):** We acknowledge that open-sourcing this work involves significant risks, particularly the potential for misuse by malicious actors. For example, certain AI product providers might exploit this technique to deliver intentionally biased LLM outputs to their users, bypassing broader societal discussions on the fairness of such practices. Nevertheless, we believe that open access is essential for the scientific community to responsibly monitor and improve these technologies, especially as similar tools are already being developed in opaque settings. By making this research publicly available, we aim to promote transparency, empowering responsible researchers to study these technologies and mitigate potential harms. We have also expanded our ethics statement to address these considerations.
>
> > **Weakness 4: Although limitations mention that a U.S.-centric focus may reduce relevance in global contexts, the topic of political discourse and presence of strongly opposing views differs in many countries. As a result, the scope is limited; also, it’s uncertain if LLMs will adapt as accurately to culturally distinct perspectives (such as those in South Asia) due to data limitations.**
>
> We thank the reviewer for highlighting this important limitation. Existing literature on political biases of LLMs, including our paper, has relatively focused on the U.S. political landscape (Santurkar et al., 2023; Motoki et al., 2024; Martin, 2023; Potter et al., 2024; Liu et al., 2022; Bang et al., 2024). We agree that data limitations concerning non-U.S. contexts may hinder LLMs' ability to accurately represent the political landscapes of other nations, where political discourse and ideological distributions are often more nuanced and complex.
>
> As shown in Appendix A.3, to partially address this concern, we conducted a preliminary analysis using the Manifesto Project dataset, which provides ideological labels for 411 political parties across various non-U.S. nations on a left-to-right continuum (-50 = left ~ 50 = right). For example, South Korea’s Democratic Party (더불어민주당) is labeled as -10, indicating a weakly left-leaning stance.
>
> When replicating our analyses in Section 4.4 (Evaluating Probes) using this dataset, we observed that linear probes were modestly accurate at predicting the political slant of non-U.S. parties, achieving a Spearman correlation of 0.531. This performance is significantly lower than that obtained for U.S. politicians (r = 0.870) and U.S. news media (r = 0.765), underscoring the challenges LLMs face in modeling political ideologies across different cultural and national contexts.
>
> However, the Manifesto Project dataset does not fully capture the global political landscape. We recognize that comprehensive data on political perspectives are scarce in many regions, including South Asia, limiting our ability to evaluate and improve LLMs for these settings. During the revision, we encouraged the AI research community to prioritize the creation of diverse and representative datasets that better reflect global political landscapes (see Appendix A.3).
>
> > **Weakness 5: Writing quality needs improvement, with several missing details. I’ve added these in the Questions below.**
>
> Please refer to our responses to the Questions below.

---

> ### Author Response · Authors · 2024-11-21
> **Official Comment by Authors (Part 3/6)**
>
> > **Weakness 6: From a theoretical standpoint, the methods used are derived from previous works. While the study provides an interesting new application, I still feel this work lacks novel theoretical contributions.**
>
> We agree! This paper does not introduce fundamentally new methodology. Rather, our focus instead is on tailoring recent probing methodology to understand how LLMs represent an important (arguably, one of the most important) concepts. Beyond the substantive results, which we believe will be of great interest to the ICLR community, we also believe this work contributes by providing a clear demonstration of recent methodology.
>
> > **Question 1. I think more details should be added in Section 3.2, Intervention. It is very unclear.**
>
> Please refer to our point-by-point responses. Additionally, we have made substantial revisions to the draft to enhance its clarity and presentation. For details, see Sections 4 and 5 of the revised draft.
>
>   - **Comments:**
>   - **In line 282,  $\theta_{l,h}$ is described as a steering vector, but in line 228, it is referred to as attention coefficients. Are they the same? If not, why are the notations the same?**
>
> Yes, they are the same.
>
> $\theta_{l,h}$ represents the regression coefficients obtained from ridge regression models during the probing phase. Specifically, for each attention head $h$ in layer $l$, we train a **separate** ridge regression model where the input is the 128-dimensional activation output of that head, and the target is the political slant. After training, $\theta_{l,h}$ is a 128-dimensional vector that captures the relationship between attention head activations and political slant. See Section 4.2 and 4.3 for details.
>
> During the intervention phase, $\theta_{l,h}$ is reused as a steering vector, allowing us to adjust the output of the corresponding attention head to simulate specific ideological perspectives. See Section 5.1 for details.
>
>   - **If $\theta_{l,h}$ represents attention coefficients from the Ridge regression model, how can it be applied directly in the attention mechanism? Are the shapes of both vectors/tensors the same? Also, as they originate from different areas, how are they connected?**
>
> Yes, the shapes of both vectors are 128-dimensional (for Llama-2-7b-chat, Mistral-7b-instruct, and Vicuna-7b).
>
> For each attention head $h$ in layer $l$, we train a **separate** ridge regression model where the input is the 128-dimensional activation output of that head, and the target is the political slant. $\theta_{l,h}$ represents the regression coefficients independently obtained from each ridge regression model during the probing phase. The shape of $\theta_{l,h}$ matches the 128-dimensional activation outputs of each attention head, ensuring compatibility.
>
> For example, Llama-2-7b-chat consists of 32 layers, each containing 32 attention heads, resulting in a total of 1,024 attention heads (32 × 32). During the probing phase, we independently train 1,024 ridge regression models, one for each attention head, producing 1,024 distinct vectors for $\theta_{l,h}$ across all attention heads. $\theta_{l,h}$ is 128-dimensional.
>
> During intervention, the same $\theta_{l,h}​$ acts as a "steering vector" to adjust the outputs of the corresponding attention head $h$ in layer $l$ by adding or subtracting a scaled version, where $\alpha$ controls the adjustment strength and $\sigma_{l,h}​$ normalizes for variance across heads. This setup connects the ridge regression-derived coefficients to the attention mechanism seamlessly. See Section 5.1 for details.
>
>   - **If $\theta_{l,h}$ is not the attention coefficient in line 282, how is it calculated? How do we maintain its shape or connect it with other components?**
>
> $\theta_{l,h}$ is the coefficient of ridge regressions where the input is attention head output of attention $h$ in layer $l$. It is re-used as a steering vector during the intervention. See our responses above.
>
>   - **In line 288, the definition, purpose, and use of $\sigma_{l,h}​$ are also unclear.**
>
> The term $\sigma_{l,h}​$ represents the standard deviation of activations for a given attention head $h$ in layer $l$. Its purpose is to normalize the intervention strength across different attention heads, as activation magnitudes can vary significantly between heads. By multiplying the steering vector $\theta_{l,h}$ ​by $\sigma_{l,h}$, we ensure that the intervention scale is consistent, preventing disproportionate influence from attention heads with higher or lower activation variance. This approach is adapted from prior work (see Li et al., 2023). See Section 5.1 for details.

---

> ### Author Response · Authors · 2024-11-21
> **Official Comment by Authors (Part 4/6)**
>
> - **In line 291, it states, We target the K most predictive attention heads for the intervention. What is the value of K here? Is it selected from the coefficients of the Ridge regression?**
>
> In line 291, $K$ refers to the number of attention heads selected for intervention. The value of $K$ is a hyperparameter, and we experiment with a predefined set of values (16, 32, 48, 64, 80, 96) to assess the impact of the number of attention heads receiving interventions.
>
> The selection of the $K$-most predictive heads is based on the performance of the ridge regression models independently trained on individual attention heads. Specifically, the heads are ranked by their predictive power, which is measured using Spearman's correlation between the predictions ($\hat{y}$ in Section 4.4) and the target labels (political slant of U.S. politicians) during cross-validation. The top $K$ heads with the highest predictive performance are then chosen for intervention. Thus, $K$ is not directly derived from the ridge regression coefficients but rather from the performance evaluation of the trained ridge regression models.
>
> For example, if $K$ = 3, the most predictive heads in Llama-2-7b-chat are Layer 15, Head 18 (r = 0.853), Layer 16, Head 11 (r = 0.845), and Layer 18, Head 4 (r = 0.844). The intervention is then applied exclusively to these heads. This approach is also adapted from prior work (see Li et al., 2023). See Section 5.1 and 5.2 for details.
>
> - **Asks:**
> - **Clearly define $\theta_{l,h}$ and explain how it relates to both the ridge regression model and the intervention process.**
>
> We define $\theta_{l,h}$ as the regression coefficients obtained from a ridge regression model during the probing phase. These coefficients are learned for each attention head $h$ in layer $l$ by separately training the ridge regression model to predict a target variable (e.g., political slant) from the activations of the attention head.
>
> During the intervention, each different $\theta_{l,h}$ is used to modify a particular attention head $h$ in layer $l$ by adding or subtracting a scaled version of $\theta_{l,h}$, thereby steering the model's output toward the left or the right. See Section 5.1 for details.
>
>   - **Explain the process of applying the ridge regression coefficients to the attention mechanism, including any necessary transformations.**
>
> The process of applying the ridge regression coefficients $\theta_{l,h}$ to the attention mechanism involves the following steps:
>
>   - Probing Phase (Learning $\theta_{l,h}$):
>     - Ridge regression models are trained separately for each attention head $h$ in each layer $l$ of the LLM.
>     - The input to the ridge regression model is the 128-dimensional activation vector from an attention head. The target variable is the political slant associated with the input context (e.g., politician, news media).
>     - The ridge regression learns a 128-dimensional coefficient vector $\theta_{l,h}$, which represents the relationship between the activations of that attention head and the political slant.
>   - Intervention Phase (Using $\theta_{l,h}$):
>     - During intervention, the learned $\theta_{l,h}$ is applied directly (without any transformation) to the activations of the corresponding attention head to steer the model's outputs (See Formula (5) at Section 5.1).
>     - $\alpha$ is a scaling parameter controlling the strength of the intervention.
>
>   - **Provide a clear definition and explanation for $\sigma_{l,h}$.**
>
>   - Purpose of Normalization ($\sigma_{l,h}$):
>     - The variance of activations can vary significantly across attention heads, which could lead to inconsistent intervention effects.
>     - The term $\sigma_{\ell,h}$ represents the standard deviation of the extracted activations ($\boldsymbol{x}^{(i)}_{\ell,h}$) from the LLM when prompted to simulate U.S. politicians in the probe training data, for a given attention head $h$ in layer $\ell$ (see Section 4.2).
>     - Its purpose is to normalize the intervention strength across different attention heads, as activation magnitudes can vary significantly between heads. By multiplying the steering vector $\theta_{l,h}$ ​by $\sigma_{l,h}$, we ensure that the intervention scale is consistent, preventing disproportionate influence from attention heads with higher or lower activation variance. This approach is adapted from prior work (see Li et al., 2023).
>
>   - **Specify how K is determined and its relationship to the ridge regression results.**
>   - Selecting Heads for Intervention:
>     - Not all attention heads are intervened upon. The $K$-most predictive heads, as determined by the ridge regression performance (i.e., Spearman correlation), are selected for intervention. The value of K is a hyperparameter, and we experiment with a predefined set of values (16, 32, 48, 64, 80, 96) to assess the impact of the number of attention heads receiving interventions, following prior research (Li et al., 2023).

---

> ### Author Response · Authors · 2024-11-21
> **Official Comment by Authors (Part 5/6)**
>
> > **Question 2: In line 223, it says, "We train a ridge regression model using U.S. politician data." Does U.S. politician data refer to the extracted activations? Is there any train-test split, or how does this part work?**
>
> Yes, the "U.S. politician data" in line 223 of the older draft refers to the extracted activations of the LLM when prompted to simulate specific U.S. politicians. To evaluate the generalizability of the ridge regression model, we use a 2-fold cross-validation strategy. First, the politicians are randomly split into two folds. In each iteration, one fold is used as the training set to fit the Ridge regression model. The other fold is used as the test set to evaluate the model's performance. We now revise Section 4.3 and Section 4.4 to clarify this.
>
> > **Question 3: In line 344, was 2-fold cross-validation done only on the Ridge regression model, or did you also collect attention heads twice?**
>
> The 2-fold cross-validation was applied to ridge regression models and the attention head activations. (However, note that attention head mechanisms in LLMs are deterministic. As long as the inputs are the same, attention head outputs should always be the same.)
>
> > **Question 4: Can you provide more details on the x-axis legends (values of each line) in Figure 3?**
>
> We trained the ridge regression models **separately** for each attention head $h$ in layer $l$. Therefore, for each attention head, the predictive performance of the corresponding linear probe (i.e., Spearman correlation coefficient) is visualized using a heatmap in Figure A2 in the revised draft (Figure 3 in the older draft).
>
> For example, Llama-2-7b-chat consists of 32 layers, each containing 32 attention heads, resulting in a total of 1,024 attention heads (32 × 32). During the probing phase, we independently trained 1,024 ridge regression models, one for each attention head. From these 1,024 models, we got 1,024 correlation coefficients. Figure A2 visualizes these 1,024 correlation coefficients.
>
> In Figure A2, each row (i.e., y-axis) represents each layer of the model from the bottom (layers close to the input layer) to the top (layers close to the output layer). Each column (i.e., x-axis) corresponds to a specific attention head in a given layer, sorted by their predictive performance in descending order of Spearman correlation. For each attention head, the predictive performance of the corresponding linear probe is visualized using a heatmap. Darker shades indicate stronger Spearman correlations, meaning the attention head was more predictive of the political slant (e.g., DW-NOMINATE or Ad Fontes Media scores). The lighter shades indicate weaker predictive performance. This visualization helps identify which layers and attention heads are most relevant for predicting ideological slant.
>
> > **Question 5: Clarify the terms accuracy and Spearman rank correlation in Section 4.1. Why is each one used?**
>
> We exclusively use Spearman rank correlation in our analysis and do not use accuracy. To avoid any potential confusion, we have revised the text to ensure consistent terminology throughout.
>
> > **Question 6: How is Figure 4 generated? Is it based on the DW-NOMINATE scores from the datasets and attention, or does it use the attention and the political aspect of the prompt?**
>
> The x-axis in Figure 2 in the revised draft (Figure 4 in the older draft) represents the predicted ideological scores ($\hat{y}_i$) generated by the ridge regression model for each entity (i.e., politicians or news media).
>
> These predicted scores are computed using linear probes applied to the attention head activations (${x}^{(i)}_{l,h}$) extracted from the LLM when it is prompted to simulate specific entities. (see Section 4.4)
>
> The y-axis represents the actual ideological scores, which are previously validated measures of political slant. For politicians, this is based on DW-NOMINATE scores, while for news media, it corresponds to Ad Fontes Media scores.

---

> ### Author Response · Authors · 2024-11-21
> **Official Comment by Authors (Part 6/6)**
>
> > **Question 7: In line 417, it says, the strongest correlation between intervention and political slant, and the previous line mentions achieving a higher degree of conservative political slant. Is the correlation value based only on steering toward a conservative slant, or on all three (liberal, conservative, and neutral)? Again, how are these values measured, and with which inputs, outputs, or parameters?**
>
> Both intervention (-30=left-30=right) and political slant (1=extreme liberal-7=extreme conservative) are continuous variables. Thus, correlation value is based on all three (liberal, conservative, and neutral).
>
> We have edited the line you mentioned to prevent the confusion. Specifically, we state:
>
> “Specifically, lower negative values of $\alpha$ resulted in outputs with a higher degree of liberal political slant, whereas higher positive values of $\alpha$ resulted in outputs with a stronger conservative slant.”
>
> Please refer to Sections 4.1, 5.1, and 5.2 in the revised draft for details on how these values are measured, as well as the corresponding inputs, outputs, and parameters.
>
> > **Question 8: What are the hyperparameters of Ridge regression model? Is it tuned? If yes, how? Is it tuned for all different situations separately, or only one model is trained - how does this work? If it is not tuned, why?**
>
> The ridge regression model used in our analysis has a single regularization hyperparameter, $\lambda$, which controls the amount of shrinkage applied to the coefficients. This hyperparameter is tuned separately for each model (i.e., Llama-2-7b-chat, Mistral-7b-instruct, Vicuna-7b) to ensure optimal performance.
>
> We perform a search over a predefined set of $\lambda$ values (0.001, 0.01, 0.1, 1, 100, 1000). The best $\lambda$ was selected based on cross-validation performance, specifically by maximizing the Spearman rank correlation between the predicted scores and the actual ideological scores of U.S. politicians (see Figure A1 and Section 4.3). As a result, we tune the $\lambda$ value to 1 across all models.
>
> > **Question 9: Adding a figure showing different parts of how the model works from start to finish, like a pipeline, could help the audience better understand the approach.**
>
> Please refer to the revised Figure A5. We have added more details to better illustrate the workflow.

---

> ### Comment · Reviewer_T21J · 2024-11-22
> **Response to Rebuttal**
>
> At first, I want to thank the authors for taking the time to prepare and share detailed responses to all my comments. I'm sorry it took six comment boxes, but I hope it helped clarify things significantly. I truly appreciate the effort and the detailed explanations you have provided.
>
> I also found the mathematical clarifications in the rebuttal comments very useful and clear. Please ensure that all of these are incorporated into the manuscript. I believe this will greatly help the audience understand the underlying mathematical processes more clearly and effectively. I also suggest including the optimal hyperparameters in the manuscript (perhaps in the Appendix) for better reproducibility.
>
> Based on the upgraded version, I have adjusted my scores as follows:
> - **Soundness:** 3 → 4
> - **Presentation:** 1 → 3
> - **Contribution:** 2 → 3
> - **Rating:** 3 → 6
>
> ---
>
> I have some additional comments and thoughts I would like you to clarify.
>
> 1. For the prompts mentioned in **A.6 (L900–927)**, are they supposed to be neutral? I feel they are not entirely factual or neutral. Also, could you please share 1–2 example responses per prompt if it’s okay to share? Additionally, connecting to my **Weakness 3**, I think it would be helpful (and beneficial for others as well) to include something like:  *This is the prompt, This is the original output, These are the steered outputs.* This way, we can clearly observe the changes and differences visually and have a better understanding.
> 2. In **Comment 6**, you state: *"This hyperparameter is tuned separately for each model (i.e., Llama-2-7b-chat, Mistral-7b-instruct, Vicuna-7b) to ensure optimal performance."*
>    However, in **Comment 3**, you mention: *"For each attention head in a layer, we train a separate ridge regression model."*
>    These statements seem a bit contradictory. Could you please clarify?

---

> > ### Author Response · Authors · 2024-11-25
> > **Official Comment by Authors (Part 2/2)**
> >
> > > **Question 2: In Comment 6, you state: "This hyperparameter is tuned separately for each model (i.e., Llama-2-7b-chat, Mistral-7b-instruct, Vicuna-7b) to ensure optimal performance."
> > However, in Comment 3, you mention: "For each attention head in a layer, we train a separate ridge regression model." These statements seem a bit contradictory. Could you please clarify?**
> >
> > We regret that our earlier description did not clearly describe the hyperparameter tuning process for ridge regressions. For each LLM, we used the same $\lambda$ across all 1,024 attention heads (32 layers × 32 heads) for simplicity. Because we intervene on the most predictive attention heads (top-$K$ heads), we optimized $\lambda$ to maximize the Spearman correlation between predicted and actual political slant among the most predictive heads. After measuring 1,024 Spearman correlation coefficients from 1,024 ridge regressions (for 1,024 heads), we optimized $r_1$ (maximum Spearman correlation from the most predictive head) and $r_{1-96}$ (average of the 96 highest correlation coefficients from the 96 most predictive heads). As shown below, $\lambda=1$ mostly optimizes these values across the three models, except for $r_{1-96}$ in Llama-2-7b (although the correlation coefficient was very close to the most optimal $\lambda$). Thus, we consistently used $\lambda=1$ across three models.
> >
> > We have included this table in the appendix (Table A6) and will provide all hyperparameters to enhance reproducibility.
> >
> > | $\lambda$ | Llama-2-7b ($r_1$) | Llama-2-7b ($r_{1-96}$) | Mistral-7b ($r_1$) | Mistral-7b ($r_{1-96}$) | Vicuna-7b ($r_1$) | Vicuna-7b ($r_{1-96}$) |
> > |-----------|--------------------|-------------------------|--------------------|-------------------------|-------------------|------------------------|
> > | 0         | 0.818              | 0.782                   | 0.815              | 0.765                   | 0.828             | 0.792                  |
> > | 0.001     | 0.835              | 0.803                   | 0.816              | 0.778                   | 0.842             | 0.812                  |
> > | 0.01      | 0.843              | 0.818                   | 0.824              | 0.789                   | 0.845             | 0.823                  |
> > | 0.1       | 0.847              | **0.823**                   | 0.839              | 0.802                   | 0.861             | 0.832                  |
> > | 1         | **0.854**              | 0.821                   | **0.846**              | **0.807**                   | **0.862**             | **0.832**                  |
> > | 100       | 0.847              | 0.782                   | 0.845              | 0.782                   | 0.852             | 0.796                  |
> > | 1000      | 0.845              | 0.775                   | 0.846              | 0.770                   | 0.846             | 0.789                  |

---

> ### Author Response · Authors · 2024-11-25
> **Official Comment by Authors (Part 1/2)**
>
> We sincerely appreciate the reviewer’s thoughtful comments and positive feedback on our revision. We have incorporated the mathematical clarifications provided in our rebuttal comments into the manuscript and will continue to ensure all necessary updates are included.
>
> **The second revised version of our paper has been uploaded for your consideration.** Below, we address the additional questions raised by the reviewer.
>
>
> > **Question1-1: For the prompts mentioned in A.6 (L900–927), are they supposed to be neutral? I feel they are not entirely factual or neutral.**
>
> We conducted additional experiments using more neutral prompts addressing the ADVANCE Act and the United Auto Workers (UAW) Strike (Both events occurred after the knowledge cutoff of Llama-2-7b-chat’s pre-training data). Specifically, after instructing ChatGPT to “remove any subjective opinions” and “extract and list the factual and neutral information in bullet points,” we constructed the following prompts:
>
> ~~~
> Write a political opinion about the following event:
> - The ADVANCE Act of 2024 was signed into law on July 9, 2024, as part of the Fire Grants and Safety Act.
> - The act supports advanced nuclear energy development in the United States.
> - It promotes generation IV nuclear reactor technology.
> - It reduces licensing costs for nuclear energy projects.
> - It extends liability protections for the nuclear industry.
> - The act directs the Nuclear Regulatory Commission (NRC) to streamline licensing processes for advanced and small modular reactors (SMRs).
> - It includes incentives for next-generation nuclear technology through reduced fees and a prize for deployment.
> - It restricts nuclear fuel imports from Russia and China.
> - It fosters U.S. nuclear exports and international collaboration.
> - The act contains provisions for environmental remediation on tribal lands.
> - Licensing changes in the act are designed to facilitate advanced reactor deployment at brownfield sites.
> - The act follows the Prohibiting Russian Uranium Imports Act.
> ~~~
>
> ~~~
> Write a political opinion about the following event:
> - The 2023 United Auto Workers (UAW) strike lasted from September 15 to October 30.
> - Approximately 49,800 union members participated in the strike.
> - The strike was directed against Ford Motor Company, General Motors, and Stellantis.
> - The primary disputes were over labor contract negotiations.
> - Key union demands included:
>   - Wage increases to counteract inflation.
>   - Elimination of a tiered employment system.
>   - Improved benefits.
>   - Worker protections against plant closures.
>   - A four-day workweek.
> - This was the first simultaneous strike against all three automakers.
> - A "rolling strike" strategy was used to conserve union resources.
> - Automakers expressed concerns about labor costs and competitiveness during the transition to electric vehicle production.
> - Tentative agreements were reached with all three companies by late October.
> - The agreements included:
>   - Significant wage increases.
>   - Reinstatement of cost-of-living adjustments.
>   - Elimination of the two-tier wage system.
> - The strike concluded after 46 days, pending union member ratification of the agreements.
> ~~~
>
>
> Essays were generated across intervention values $\alpha$ (-30, -20, -10, 0, 10, 20, 30) with varying prompt templates (e.g., Write a [political opinion/essay/statement] about the following event), resulting in 21 essays per event. $K$ was set to 96 based on our results indicating that $K=96$ produces the best intervention outcomes (Section 5.3).
>
> We find that the intervention ($\alpha$) is significantly correlated with political slant in outputs for both cases. Specifically, both the ADVANCE Act ($r=0.479$, $p=0.028$, $N=21$) and the UAW strike ($r=0.470$, $p=0.031$, $N=21$) exhibited significant correlations.
>
> Regarding the Israel-Hamas war case, we crafted the original prompt based on language from a major press outlet. However, recognizing the wide range of valid perspectives on this issue, we have decided not to include it in the second version of revision.
>
> > **Question 1-2: Also, could you please share 1–2 example responses per prompt if it’s okay to share? Additionally, connecting to my Weakness 3, I think it would be helpful (and beneficial for others as well) to include something like: This is the prompt, This is the original output, These are the steered outputs. This way, we can clearly observe the changes and differences visually and have a better understanding.**
>
>
> Tables A3 and A4 in the prior draft (first revised version) present original and steered outputs. However, we have created a new Appendix section (See Appendix A.9) to showcase more examples, including prompts, original outputs, and steered outputs for the ADVANCE Act and UAW strike cases.

---

> ### Comment · Reviewer_T21J · 2024-11-25
> **Official Comment (Response to 2nd Stage Responses)**
>
> Thank you for addressing my concerns and providing clarification on these issues. I hope this has contributed to improving the overall quality of the work. I will update the scores accordingly (6 -> 8).

---

### Official Review · Reviewer_vb63 · 2024-11-02

**Soundness:** 4
**Presentation:** 4
**Contribution:** 4
**Rating:** 10
**Confidence:** 4

**Summary:**

This paper presents an analysis of how LLM's internal activation layers capture political slant according to a US-centric liberal-conservative definition. By leveraging an automatically created dataset of texts generated by prompting LLMs with various politician personas, authors create linear probes to isolate attention heads that capture this political dimension. They show through correlation and intervention experiments that these activations can indeed capture and affect the political slant of generated texts.

**Strengths:**

This paper is very close to a slam dunk in my opinion.
- I liked the goal of the paper, i.e., to explore political leaning in LLMs' activations.
- The methodology is grounded in poliSci theories (DW-nominate)
- The correlation experiments are well set up.
- I really liked the intervention experiments.

**Weaknesses:**

There really aren't many weaknesses in this paper. All I could find were opportunities for improvement.
- I wish the paper's novelty was highlighted a little more in the intro and related work. Is this the first work that has examined the linear layers of LLMs for political bias?
- There are some missing related works particularly from the NLP side of things (e.g., https://aclanthology.org/2023.acl-long.656). I wish authors had done a little bit more comprehensive of a lit search, particularly in the ACL anthology (https://aclanthology.org) which contains papers on this topic.
- I would have liked a more nuanced discussion around political biases in LLMs. The current paper (§2) frames political biases as an issue of fairness, i.e., that LLMs that do not represent all political ideologies equally are not fair. However, there are issues that emerge from this framing, for example, the issue of false-balance / bothsideism (https://en.wikipedia.org/wiki/False_balance). Additionally, there is a growing discussion that fairness and non-bias are most important insofar as they can help shift power away from oppressive institutions (https://www.aclweb.org/anthology/2020.acl-main.485.pdf). I wish authors gave a little more balanced of a perspective on this.
- Footnote 1: I don't fully understand the distinction between "stochastic parroting" vs. linear representations, and specifically why if LLMs are merely stochastic parrots there would be no linear representation. It feels like the argument is somewhat of a side point given the results of the paper. Or it may be worth adding a longer discussion at the end of the paper about whether this means LLMs encode more systematic representations?
- L328: I really liked the intervention analyses, yet I wish that authors had given more empirical evidence that GPT-4 can indeed classify the political slant of the essays generated. This is important because LLMs can have various biases when used as text judges. The easiest method would be to conduct a small-scale in-house manual validation, where authors or colleagues rate the political slant of the generated texts and compare their ratings to the GPT-4 scores. I suspect the alignment is high but it would be best to confirm. Alternatively, there are other trained classifiers out there to predict the political leaning of text that authors could use.
- L429: I wish authors had contextualized the claim that conservative arguments might just be shorter, grounded in evidence from political science to back up this hypothesis. This is an interesting note, and may point to various things like tendency to over-simplify arguments on side of the political spectrum, to resort to emotional and persuasive talking points instead of more rational ones, etc. I wish authors had discussed this in further depth.

**Questions:**

This is maybe a weird question, but I'm curious why authors decided to submit to ICLR instead of ACL venues (ACL rolling review), given the very NLP-nature of this work? I'm particularly curious because the field boundaries between ML and NLP are changing rapidly with the rise of LLMs, and I would just like to know. It feels like this work would be really well received in NLP conferences.

---

> ### Author Response · Authors · 2024-11-21
> **Official Comment by Authors (Part 1/2)**
>
> We thank Reviewer vb63 for their thoughtful engagement and valuable feedback on our manuscript. We have made substantial revisions and **submitted an updated version of the manuscript** in accord with these expressed concerns and constructive suggestions.
>
> > **Weakness 1: I wish the paper's novelty was highlighted a little more in the intro and related work. Is this the first work that has examined the linear layers of LLMs for political bias?**
>
> We thank the reviewer for recognizing the novelty of our work. To the best of our knowledge, this paper is the first to systematically examine the linear representation of political bias in LLMs. We have highlighted this point in the introduction and related work (see Line 53 and Line 99).
>
> > **Weakness 2: There are some missing related works particularly from the NLP side of things (e.g., https://aclanthology.org/2023.acl-long.656). I wish authors had done a little bit more comprehensive of a lit search, particularly in the ACL anthology (https://aclanthology.org) which contains papers on this topic.**
>
> We have incorporated additional related works from the NLP community in the revised draft, including papers from the ACL Anthology. Specifically, we added references on measuring political biases in LLMs (Feng et al., 2023; Bang et al., 2024), how political biases in pre-training corpora can manifest in downstream tasks (Feng et al., 2023; Jiang et al., 2022), the political prudence of chatbots (Bang et al., 2021), and the challenges of measuring political biases that manifest in open-ended responses using close-ended survey questions (Goldfarb-Tarrant et al., 2021; Röttger et al., 2024). See Section 2: Political Bias of LLMs.
>
> > **Weakness 3: I would have liked a more nuanced discussion around political biases in LLMs. The current paper (§2) frames political biases as an issue of fairness, i.e., that LLMs that do not represent all political ideologies equally are not fair. However, there are issues that emerge from this framing, for example, the issue of false-balance / bothsideism (https://en.wikipedia.org/wiki/False_balance). Additionally, there is a growing discussion that fairness and non-bias are most important insofar as they can help shift power away from oppressive institutions (https://www.aclweb.org/anthology/2020.acl-main.485.pdf). I wish authors gave a little more balanced of a perspective on this.**
>
> We agree that balance and fairness are not synonymous. In response, we have now revised the text to reflect diverse perspectives on fairness. See updated footnote 1 on page 3 of the revised draft. Specifically, we now state:
>
> “We note that political balance and fairness are not synonymous. There are diverse views on how to ensure fairness in LLMs concerning political bias. Some advocate for representing a wide range of political perspectives as a form of fairness (Sorensen et al., 2024), while others emphasize that fairness is most important insofar as it shifts power away from oppressive institutions in favor of underrepresented stances and perspectives (Blodgett et al., 2020).”
>
> > **Weakness 4: Footnote 1: I don't fully understand the distinction between "stochastic parroting" vs. linear representations, and specifically why if LLMs are merely stochastic parrots there would be no linear representation. It feels like the argument is somewhat of a side point given the results of the paper. Or it may be worth adding a longer discussion at the end of the paper about whether this means LLMs encode more systematic representations?**
>
> We recognize that footnote 1 in the original submission may have detracted from the main argument of our paper, so we have removed it.
>
> Our intention was to address the following: The original “stochastic parroting” paper raised the concern that “LLMs may learn a massive collection of correlations but lack any coherent model or understanding of the underlying data-generating process given text-only training” (Bender et al., 2021). Recent work on linear representations of high-level concepts, however, including our own, suggests that this may not always be the case (i.e., LLMs learn parsimonious, linear representations underlying text generation) (Gurnee and Tegmark, 2022). We believe this discussion fits more naturally in Lines 97–103 of the revised draft and have accordingly removed the footnote.

---

> ### Author Response · Authors · 2024-11-21
> **Official Comment by Authors (Part 2/2)**
>
> > **Weakness 5: L328: I really liked the intervention analyses, yet I wish that authors had given more empirical evidence that GPT-4 can indeed classify the political slant of the essays generated. This is important because LLMs can have various biases when used as text judges. The easiest method would be to conduct a small-scale in-house manual validation, where authors or colleagues rate the political slant of the generated texts and compare their ratings to the GPT-4 scores. I suspect the alignment is high but it would be best to confirm. Alternatively, there are other trained classifiers out there to predict the political leaning of text that authors could use.**
>
> We thought this was a great idea and have run a survey-based validation, which is now reported in the revised draft. We validated GPT-4’s evaluations against a politically balanced sample of human annotators. Using the CloudResearch survey platform, we recruited a sample of 10 U.S. residents—comprising 3 Democrats, 4 Independents, and 3 Republicans—to annotate a random selection of 21 essays generated by Llama-2-7b-chat (see Appendix A.2.3). After averaging the scores from these human annotators, we measured the inter-rater reliability between GPT-4 and the human average.
>
> As anticipated by the Reviewer, we found very high inter-rater reliability, indicating strong alignment between GPT-4 and the average human annotator scores ($ICC(A,1) = 0.91$), which supports the validity of GPT-4 in annotating political slant. Additionally, the Spearman correlation between GPT-4 and the average human scores was also very high ($r = 0.952, p < 10^{-10}$).
>
> > **Weakness 6: L429: I wish authors had contextualized the claim that conservative arguments might just be shorter, grounded in evidence from political science to back up this hypothesis. This is an interesting note, and may point to various things like tendency to over-simplify arguments on side of the political spectrum, to resort to emotional and persuasive talking points instead of more rational ones, etc. I wish authors had discussed this in further depth.**
>
> We appreciate the reviewer’s suggestion to provide more context on why conservative arguments tend to be shorter on specific topics, such as gun control. As the reviewer noted, research has shown that conservatives and liberals often utilize different persuasive strategies, with conservatives leaning towards more intuitive, emotionally resonant arguments (e.g., Cakanlar and White, 2023). We also recognize that underlying ideological differences, such as distinct moral foundations—where conservatives may prioritize values like "liberty" and liberals emphasize "care"—could contribute to variations in argument length (e.g., Graham, Haidt, and Nosek, 2009). This is a very intriguing hypothesis generated by our model, which deserves focused follow-up research for further validation. Therefore, we have briefly addressed these potential influences in Footnote 14 and suggested directions for future research to explore these complexities further.
>
> > **Question 1: This is maybe a weird question, but I'm curious why authors decided to submit to ICLR instead of ACL venues (ACL rolling review), given the very NLP-nature of this work? I'm particularly curious because the field boundaries between ML and NLP are changing rapidly with the rise of LLMs, and I would just like to know. It feels like this work would be really well received in NLP conferences.**
>
> We did consider submitting to an NLP venue! As you mentioned, our work is at the intersection of two communities. It speaks to the NLP community on political bias in LLMs (e.g., Santurkar et al., ICML 2023), and it also speaks to the ML community focused on probing transformers and understanding representations in LLMs. It was a close decision, but we felt that the ICLR audience would appreciate the work as a clear demonstration of recent probing methods developed by other ICLR papers, such as Gurnee and Tegmark, ICLR 2023.

---

> ### Comment · Reviewer_vb63 · 2024-11-21
>
> Thanks for the response! My score remains (equally positive) :)

---

> > ### Author Response · Authors · 2024-11-21
> >
> > Thank you for the positive assessment! We really appreciate your thoughtful and constructive feedback.

---

### Official Review · Reviewer_uBqm · 2024-11-12

**Soundness:** 3
**Presentation:** 3
**Contribution:** 3
**Rating:** 6
**Confidence:** 4

**Summary:**

The authors identify a set of attention heads within select large language models (LLMs) that encode ideological perspectives. Through the use of linear probes on the model's layer representations, they uncover heads associated with ideological slant. Their findings demonstrate that modifying these neurons influences the ideological bias exhibited by the model.

**Strengths:**

* The authors find a set of attention heads encoding political slant in 3 LLMs
* The authors causally ablate these heads and show the resulting effects on text generation

**Weaknesses:**

*  Why was ridge regression used instead of standard linear regression? Were collinearities established prior to making this choice?
* The authors don't establish baseline behaviours -- or maybe I am missing this. What were the model's ideological leanings before the intervention was applied?

**Questions:**

N/A

**Details Of Ethics Concerns:**

The paper investigates steering the political ideologies of LLMs, though it doesn't take a political stance. The paper can be used to make an LLM more or less liberal for example, and this can be dangerous.

---

> ### Author Response · Authors · 2024-11-21
> **Official Comment by Authors**
>
> We thank Reviewer uBqm for their thoughtful engagement and valuable feedback on our manuscript. We have made substantial revisions and **submitted an updated version of the manuscript** in accord with these expressed concerns and constructive suggestions.
>
> > **Weakness 1: Why was ridge regression used instead of standard linear regression? Were collinearities established prior to making this choice?**
>
> We used ridge regression to address overfitting and improve generalization, particularly in the presence of collinearity among features. As the reviewer noted, the features (i.e., neuron activations within each attention head) exhibited significant collinearity. For example, in Llama-2-7b-chat, variance inflation factor (VIF) values exceeded 10 for 807 out of 1,024 attention heads (78.8%), indicating high collinearity (see Footnote 5).
>
> Additionally, ridge regression has been widely adopted in similar contexts to analyze high-dimensional features in LLMs. For example, Gurnee and Tegmark (2023) utilized ridge regression to investigate high-level concepts (i.e., time and space) in LLMs.
>
> > **Weakness 2: The authors don't establish baseline behaviours -- or maybe I am missing this. What were the model's ideological leanings before the intervention was applied?**
>
> Figure 3a in the revised draft (Figure 4a in the older draft) actually reports baselines, which corresponds to $\alpha=0$, but we have revised the paper to highlight the baseline behaviors (see Section 5.3. Results). We have now highlighted that the average political slant at  $\alpha = 0$ was consistently below 4 (on a scale of 1 = extreme liberal to 7 = extreme conservative), indicating a default left-leaning bias across all models. In other words, without any intervention, the models' outputs tend to align more closely with left-leaning political opinions. For example:
>
>   - **Llama-2-7b-chat**: Average slant = 2.296, Standard Deviation = 1.222
>   - **Mistral-7b-instruct**: Average slant = 2.778, Standard Deviation = 0.925
>   - **Vicuna-7b**: Average slant = 2.685, Standard Deviation = 1.669
>
> This observed baseline behavior aligns with prior research, which has documented similar left-leaning political tendencies in LLMs (e.g., Santurkar et al., 2023; Motoki et al., 2024; Martin, 2023; Potter et al., 2024; Liu et al., 2022; Bang et al., 2024).

---

> > ### Comment · Reviewer_uBqm · 2024-11-25
> >
> > Thanks for the clarifications!

---

### Author Response · Authors · 2024-11-21

We are grateful for the reviewers’ thoughtful engagement with our paper and their constructive feedback. **In response to the reviewers' comments, we have made substantial revisions and submitted an updated version of the manuscript.** Below, we address each reviewer’s comments individually.

---

### Meta-Review · Area_Chair_yXyg · 2024-12-21

**Metareview:**

**Summary:**

In this paper, the authors seek to isolate which transformer LM hidden layers are most highly influential over political bias in outputs and identify attention heads that can represent this bias based on correlation with previously proposed political stance scoring methodologies. Using causal interventions, they show that post-hoc manipulations of transformer LMs can modify their political alignment without training or sophisticated feature-engineering.

**Strengths:**

- These are very strong findings highlighting not only that transformer LMs are capable of abstract representation of political ideologies, but that these representations can be controlled through causal intervention.  Results confirm this approach is effective even for generations involving post-model knowledge cutoff events.

- Their political ideology probing methodology, prompting LMs to imitate speech of known political figures or news media, is innovative and well-grounded.

**Weaknesses:**

 - My primary concern would be the accuracy of GPT-4 as a judge in evaluating political ideology, but they show that GPT-4 performs relatively well at evaluating political ideology.

I believe this paper nicely builds on prior linear probing literature while providing a novel perspective on how LLMs encode political stances. Their proposed method for controlling political bias could be broadly useful to the research community. From my own reading and the authors' rebuttal, I believe this is a high-quality paper and see no reason not to accept.

**Additional Comments On Reviewer Discussion:**

The reviewers are unanimously supporting acceptance. Primary concerns were the lack of clarity around the authors' definition of linear representation, choice of ridge regression and missing baseline ideology measurements, but these concerns have now been addressed in the revised draft. I would encourage more discussion of how models’ left-leaning bias influences intervention effectiveness.

---

### Decision · Program_Chairs · 2025-01-22

Accept (Oral)